# Finding and Listing Front-door Adjustment Sets

**Hyunchai Jeong**
Purdue University
jeong3@purdue.edu

**Jin Tian**
Iowa State University
jtian@iastate.edu

**Elias Bareinboim**
Columbia University
eb@cs.columbia.edu

## Abstract

Identifying the effects of new interventions from data is a significant challenge found across a wide range of the empirical sciences. A well-known strategy for identifying such effects is Pearl's *front-door (FD) criterion* [26]. The definition of the FD criterion is declarative, only allowing one to decide whether a specific set satisfies the criterion. In this paper, we present algorithms for finding and enumerating possible sets satisfying the FD criterion in a given causal diagram. These results are useful in facilitating the practical applications of the FD criterion for causal effects estimation and helping scientists to select estimands with desired properties, e.g., based on cost, feasibility of measurement, or statistical power.

## 1 Introduction

Learning cause and effect relationships is a fundamental challenge across data-driven fields. For example, health scientists developing a treatment for curing lung cancer need to understand how a new drug affects the patient's body and the tumor's progression. The distillation of causal relations is indispensable to understanding the dynamics of the underlying system and how to perform decision-making in a principled and systematic fashion [27, 37, 2, 30, 1, 23, 24].

One of the most common methods for learning causal relations is through *Randomized Controlled Trials* (RCTs, for short) [8]. RCTs are considered as the "gold standard" in many fields of empirical research and are used throughout the health and social sciences as well as machine learning and AI. In practice, however, RCTs are often hard to perform due to ethical, financial, and technical issues. For instance, it may be unethical to submit an individual to a certain condition if such condition may have some potentially negative effects (e.g., smoking). Whenever RCTs cannot be conducted, one needs to resort to analytical methods to infer causal relations from observational data, which appears in the literature as the problem of *causal effect identification* [26, 27].

The causal identification problem asks whether the effect of holding a variable $X$ at a constant value $x$ on a variable $Y$, written as $P(Y|do(X = x))$, or $P(Y|do(x))$, can be computed from a combination of observational data and causal assumptions. One of the most common ways of eliciting these assumptions is in the form of a causal diagram represented by a directed acyclic graph (DAG), where its nodes and edges describe the underlying data generating process. For instance, in Fig. 1a, three nodes $X, Z, Y$ represent variables, a directed edge $X \to Z$ indicates that $X$ causes $Z$, and a dashed-bidirected edge $X \leftrightarrow Y$ represents that $X$ and $Y$ are confounded by unmeasured (latent) factors. Different methods can solve the identification problem, including Pearl's celebrated do-calculus [26] as well as different algorithmic solutions [40, 34, 12].

In practice, researchers often rely on identification strategies that generate well-known identification formulas. One of the arguably most popular strategies is identification by covariate adjustment. Whenever a set $Z$ satisfies the *back-door (BD) criterion* [26] relative to the pair $X$ and $Y$, where $X$ and $Y$ represent the treatment and outcome variables, respectively, the causal effect $P(Y|do(x))$ can be evaluated through the BD adjustment formula $\sum_z P(y|x, z)P(z)$.

36th Conference on Neural Information Processing Systems (NeurIPS 2022).

Despite the popularity of the covariate adjustment technique for estimating causal effects, there are still settings in which no BD admissible set exists. For example, consider the causal diagram $\mathcal{G}$ in Fig. 1a. There clearly exists no set to block the BD path from $X$ to $Y$, through the bidirected arrow, $X \leftrightarrow Y$. One may surmise that this effect is not identifiable and the only one of evaluating the interventional distribution is through experimentation. Still, this is not the case. The effect $P(Y|do(x))$ is identifiable from $\mathcal{G}$ and the observed distribution $P(x, y, z)$ over $\{X, Y, Z\}$ by another classic identification strategy known as the *front-door (FD) criterion* [26]. In particular, through the following FD adjustment formula provides the way of evaluating the interventional distribution:

$$P(Y|do(x)) = \sum_z P(z|x) \sum_{x'} P(y|x', z)P(x'). \tag{1}$$

We refer to Pearl and Mackenzie [28, Sec. 3.4] for an interesting account of the history of the FD criterion, which was the first graphical generalization of the BD case. The FD criterion is drawing more attention in recent years. For applications of the FD criterion, see, e.g., Hünermund and Bareinboim [13] and Glynn and Kashin [10]. Statistically efficient and doubly robust estimators have recently been developed for estimating the FD estimand in Eq. (1) from finite samples [9], which are still elusive for arbitrary estimands identifiable in a diagram despite recent progress [18, 19, 5, 20, 43].

Both the BD and FD criteria are only descriptive, i.e., they specify whether a specific set $Z$ satisfies the criteria or not, but do not provide a way to find an admissible set $Z$. In addition, in many situations, it is possible that multiple adjustment sets exist. Consider for example the causal diagram in Fig. 1b, and the task of identifying the effect of $X$ on $Y$. The distribution $P(Y|do(x))$ can indeed be identified by the FD criterion with a set $Z = \{A, B, C\}$ given by the expression in Eq. (1) (with $Z$ replaced with $\{A, B, C\}$). Still, what if the variable $B$ is costly to measure or encodes some personal information about patients which is undesirable to be shared due to ethical concerns? In this case, the set $Z = \{A, C\}$ also satisfies the FD criterion and may be used. Even when both $B$ and $C$ are unmeasured, the set $Z = \{A\}$ is also FD admissible.

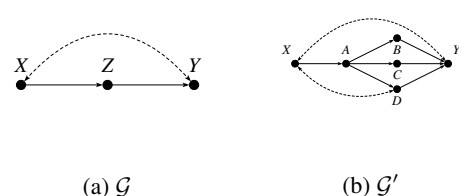

(a) $\mathcal{G}$        (b) $\mathcal{G}'$

Figure 1: (a) A canonical example of the FD criterion where $\{Z\}$ satisfies the FD criterion relative to $(\{X\}, \{Y\})$. In (b), four FD adjustment sets relative to $(\{X\}, \{Y\})$ are available: $\{A\}$, $\{A, B\}$, $\{A, C\}$, and $\{A, B, C\}$.

This simple example shows that a target effect can be estimated using different adjustment sets leading to different probability expressions over different set of variables, which has important practical implications. Each variable implies different practical challenges in terms of measurement, such as cost, availability, privacy. Each estimand has different statistical properties in terms of sample complexity, variance, which may play a key role in the study design [31, 11, 32, 36]. Algorithms for finding and listing all possible adjustment sets are hence very useful in practice, which will allow scientists to select an adjustment set that exhibits desirable properties. Indeed, algorithms have been developed in recent years for finding one or listing all BD admissible sets [38, 39, 41, 29, 42]. However, no such algorithm is currently available for finding/listing FD admissible sets.

The goal of this paper is to close this gap to facilitate the practical applications of the FD criterion for causal effects estimation and help scientists to select estimand with certain desired properties [1]. Specifically, the contributions of this paper are as follows:

1. We develop an algorithm that finds an admissible front-door adjustment set $\mathbf{Z}$ in a given causal diagram in polynomial time (if one exists). We solve a variant of the problem that imposes constraints $\mathbf{I} \subseteq \mathbf{Z} \subseteq \mathbf{R}$ for given sets $\mathbf{I}$ and $\mathbf{R}$, which allows a scientist to constrain the search to include specific subsets of variables or exclude variables from search perhaps due to cost, availability, or other technical considerations.

2. We develop a sound and complete algorithm that enumerates all front-door adjustment sets with polynomial delay - the algorithm takes polynomial amount of time to return each new admissible set, if one exists, or return failure whenever it exhausted all admissible sets.

---

[1]Code is available at `https://github.com/CausalAILab/FrontdoorAdjustmentSets`.

## 2 Preliminaries

**Notation.** We write a variable in capital letters ($X$) and its value as small letters ($x$). Bold letters, $\mathbf{X}$ or $\mathbf{x}$, represent a set of variables or values. We use kinship terminology to denote various relationships in a graph $\mathcal{G}$ and denote the parents, ancestors, and descendants of $\mathbf{X}$ (including $\mathbf{X}$ itself) as $Pa(\mathbf{X})$, $An(\mathbf{X})$, and $De(\mathbf{X})$, respectively. Given a graph $\mathcal{G}$ over a set of variables $\mathbf{V}$, a subgraph $\mathcal{G}_{\mathbf{X}}$ consists of a subset of variables $\mathbf{X} \subseteq \mathbf{V}$ and their incident edges in $\mathcal{G}$. A graph $\mathcal{G}$ can be transformed: $\mathcal{G}_{\overline{\mathbf{X}}}$ is the graph resulting from removing all incoming edges to $\mathbf{X}$, and $\mathcal{G}_{\underline{\mathbf{X}}}$ is the graph with all outgoing edges from $\mathbf{X}$ removed. A DAG $\mathcal{G}$ may be *moralized* into an undirected graph where all directed edges of $\mathcal{G}$ are converted into undirected edges, and for every pair of nonadjacent nodes in $\mathcal{G}$ that share a common child, an undirected edge that connects such pair is added [22].

A path $\pi$ from a node $X$ to a node $Y$ in $\mathcal{G}$ is a sequence of edges where $X$ and $Y$ are the endpoints of $\pi$. A node $W$ on $\pi$ is said to be a collider if $W$ has converging arrows into $W$ in $\pi$, e.g., $\rightarrow W \leftarrow$ or $\leftrightarrow W \leftarrow$. $\pi$ is said to be blocked by a set $\mathbf{Z}$ if there exists a node $W$ on $\pi$ satisfying one of the following two conditions: 1) $W$ is a collider, and neither $W$ nor any of its descendants are in $\mathbf{Z}$, or 2) $W$ is not a collider, and $W$ is in $\mathbf{Z}$ [25]. Given three disjoint sets $\mathbf{X}, \mathbf{Y}$, and $\mathbf{Z}$ in $\mathcal{G}$, $\mathbf{Z}$ is said to $d$-separate $\mathbf{X}$ from $\mathbf{Y}$ in $\mathcal{G}$ if and only if $\mathbf{Z}$ blocks every path from a node in $\mathbf{X}$ to a node in $\mathbf{Y}$ according to the $d$-separation criterion [25], and we say that $\mathbf{Z}$ is a *separator* of $\mathbf{X}$ and $\mathbf{Y}$ in $\mathcal{G}$.

**Structural Causal Models (SCMs).** We use Structural Causal Models (SCMs, for short) [27] as our basic semantical framework. An SCM is a 4-tuple $\langle \mathbf{U}, \mathbf{V}, \mathbf{F}, P(\mathbf{u}) \rangle$, where 1) $\mathbf{U}$ is a set of exogenous (latent) variables, 2) $\mathbf{V}$ is a set of endogenous (observed) variables, 3) $\mathbf{F}$ is a set of functions $\{f_V\}_{V \in \mathbf{V}}$ that determine the value of endogenous variables, e.g., $v \leftarrow f_V(\mathbf{pa}_V, \mathbf{u}_V)$ is a function with $\mathbf{PA}_V \subseteq \mathbf{V} \setminus \{V\}$ and $\mathbf{U}_V \subseteq \mathbf{U}$, and 4) $P(\mathbf{u})$ is a joint distribution over the exogenous variables $\mathbf{U}$. Each SCM induces a *causal diagram* $\mathcal{G}$ [3, Def. 13] where every variable $v \in \mathbf{V}$ is a vertex and directed edges in $\mathcal{G}$ correspond to functional relationships as specified in $\mathbf{F}$ and dashed bidirected edges represent common exogenous variables between two vertices. Within the structural semantics, performing an intervention and setting $X = x$ is represented through the do-operator, $do(X = x)$, which encodes the operation of replacing the original functions of $X$ (i.e., $f_X(\mathbf{pa}_X, \mathbf{u}_X)$) by the constant $x$ and induces a submodel $\mathcal{M}_x$ and an interventional distribution $P(v|do(x))$.

**Classic Causal Effects Identification Criteria.** Given a causal diagram $\mathcal{G}$ over $\mathbf{V}$, an effect $P(\mathbf{y}|do(\mathbf{x}))$ is said to be *identifiable* in $\mathcal{G}$ if $P(\mathbf{y}|do(\mathbf{x}))$ is uniquely computable from the observed distribution $P(\mathbf{v})$ in any SCM that induces $\mathcal{G}$ [27, p. 77].

A path between $X$ and $Y$ with an arrow into $X$ is known as a *back-door* path from $X$ to $Y$. The celebrated back-door (BD) criterion [26] provides a sufficient condition for effect identification from observational data, which states that if a set $\mathbf{Z}$ of non-descendants of $\mathbf{X}$ blocks all BD paths from $\mathbf{X}$ to $\mathbf{Y}$, then the causal effect $P(\mathbf{y}|do(\mathbf{x}))$ is identified by the BD adjustment formula:

$$P(\mathbf{y}|do(\mathbf{x})) = \sum_{\mathbf{z}} P(\mathbf{y}|\mathbf{x}, \mathbf{z})P(\mathbf{z}) \tag{2}$$

Another classic identification condition that is key to the discussion in this paper is known as the front-door criterion, which is defined as follows:

**Definition 1.** (Front-door (FD) Criterion [26]) A set of variables $\mathbf{Z}$ is said to satisfy the front-door criterion relative to the pair $(\mathbf{X}, \mathbf{Y})$ if

1. $\mathbf{Z}$ intercepts all directed paths from $\mathbf{X}$ to $\mathbf{Y}$,

2. There is no unblocked back-door path from $\mathbf{X}$ to $\mathbf{Z}$, and

3. All back-door paths from $\mathbf{Z}$ to $\mathbf{Y}$ are blocked by $\mathbf{X}$, i.e., $\mathbf{X}$ is a separator of $\mathbf{Z}$ and $\mathbf{Y}$ in $\mathcal{G}_{\underline{\mathbf{Z}}}$.

If $\mathbf{Z}$ satisfies the FD criterion relative to the pair $(\mathbf{X}, \mathbf{Y})$, then $P(\mathbf{y}|do(\mathbf{x}))$ is identified by the following FD adjustment formula [26]:

$$P(\mathbf{y}|do(\mathbf{x})) = \sum_{\mathbf{z}} P(\mathbf{z}|\mathbf{x}) \sum_{\mathbf{x}'} P(\mathbf{y}|\mathbf{x}', \mathbf{z})P(\mathbf{x}'). \tag{3}$$

# 3 Finding A Front-door Adjustment Set

In this section, we address the following question: given a causal diagram $\mathcal{G}$, is there a set $\mathbf{Z}$ that satisfies the FD criterion relative to the pair $(\mathbf{X}, \mathbf{Y})$ and, therefore, allows us to identify $P(\mathbf{y}|do(\mathbf{x}))$ by the FD adjustment? We solve a more general variant of this question that imposes a constraint $\mathbf{I} \subseteq \mathbf{Z} \subseteq \mathbf{R}$ for given sets $\mathbf{I}$ and $\mathbf{R}$. Here, $\mathbf{I}$ are variables that must be included in $\mathbf{Z}$ ($\mathbf{I}$ could be empty) and $\mathbf{R}$ are variables that could be included in $\mathbf{Z}$ ($\mathbf{R}$ could be $\mathbf{V} \backslash (\mathbf{X} \cup \mathbf{Y})$). Note the constraint that variables in $\mathbf{W}$ *cannot* be included can be enforced by excluding $\mathbf{W}$ from $\mathbf{R}$. Solving this version of the problem will allow scientists to put constraints on candidate adjustment sets based on practical considerations. In addition, this version will form a building block for an algorithm that enumerates all FD admissible sets in a given $\mathcal{G}$ - the algorithm LISTFDSETS (shown in Alg. 2 in Section 4) for listing all FD admissible sets will utilize this result during the recursive call.

---

**Algorithm 1** FINDFDSET $(\mathcal{G}, \mathbf{X}, \mathbf{Y}, \mathbf{I}, \mathbf{R})$

---

1: **Input:** $\mathcal{G}$ a causal diagram; $\mathbf{X}, \mathbf{Y}$ disjoint sets of variables; $\mathbf{I}, \mathbf{R}$ sets of variables.
2: **Output:** $\mathbf{Z}$ a set of variables satisfying the front-door criterion relative to $(\mathbf{X}, \mathbf{Y})$ with the constraint $\mathbf{I} \subseteq \mathbf{Z} \subseteq \mathbf{R}$.
3: **Step 1:**
4:     $\mathbf{R}' \leftarrow$ GETCAND2NDFDC $(\mathcal{G}, \mathbf{X}, \mathbf{I}, \mathbf{R})$
5:     **if** $\mathbf{R}' = \perp$ **then: return** $\perp$
6: **Step 2:**
7:     $\mathbf{R}'' \leftarrow$ GETCAND3RDFDC $(\mathcal{G}, \mathbf{X}, \mathbf{Y}, \mathbf{I}, \mathbf{R}')$
8:     **if** $\mathbf{R}'' = \perp$ **then: return** $\perp$
9: **Step 3:**
10:     $\mathcal{G}' \leftarrow$ GETCAUSALPATHGRAPH $(\mathcal{G}, \mathbf{X}, \mathbf{Y})$
11:     **if** TESTSEP $(\mathcal{G}', \mathbf{X}, \mathbf{Y}, \mathbf{R}'') =$ True **then:**
12:         **return** $\mathbf{Z} = \mathbf{R}''$
13:     **else: return** $\perp$

---

We have developed a procedure called FINDFDSET shown in Alg. 1 that outputs a FD adjustment set $\mathbf{Z}$ relative to $(\mathbf{X}, \mathbf{Y})$ satisfying $\mathbf{I} \subseteq \mathbf{Z} \subseteq \mathbf{R}$, or outputs $\perp$ if none exists, given a causal diagram $\mathcal{G}$, disjoint sets of variables $\mathbf{X}$ and $\mathbf{Y}$, and two sets of variables $\mathbf{I}$ and $\mathbf{R}$.

**Example 1.** Consider the causal graph $\mathcal{G}'$, shown in Fig. 1b, with $\mathbf{X} = \{X\}$, $\mathbf{Y} = \{Y\}$, $\mathbf{I} = \emptyset$ and $\mathbf{R} = \{A, B, C, D\}$. Then, FINDFDSET outputs $\{A, B, C\}$. With $\mathbf{I} = \{C\}$ and $\mathbf{R} = \{A, C\}$, FINDFDSET outputs $\{A, C\}$. With $\mathbf{I} = \{D\}$ and $\mathbf{R} = \{A, B, C, D\}$, FINDFDSET outputs $\perp$ as no FD adjustment set that contains $D$ is available.

FINDFDSET runs in three major steps. Each step identifies candidate variables that incrementally satisfy each of the conditions of the FD criterion relative to $(\mathbf{X}, \mathbf{Y})$. First, FINDFDSET constructs a set of candidate variables $\mathbf{R}'$, with $\mathbf{I} \subseteq \mathbf{R}' \subseteq \mathbf{R}$, such that every subset $\mathbf{Z}$ with $\mathbf{I} \subseteq \mathbf{Z} \subseteq \mathbf{R}'$ satisfies the second condition of the FD criterion (i.e., there is no BD path from $\mathbf{X}$ to $\mathbf{Z}$). Next, FINDFDSET generates a set of candidate variables $\mathbf{R}''$, with $\mathbf{I} \subseteq \mathbf{R}'' \subseteq \mathbf{R}'$, such that for every variable $v \in \mathbf{R}''$, there exists a set $\mathbf{Z}$ with $\mathbf{I} \subseteq \mathbf{Z} \subseteq \mathbf{R}'$ and $v \in \mathbf{Z}$ that further satisfies the third condition of the FD criterion, that is, all BD paths from $\mathbf{Z}$ to $\mathbf{Y}$ are blocked by $\mathbf{X}$. Finally, FINDFDSET outputs a set $\mathbf{Z}$ that further satisfies the first condition of the FD criterion - $\mathbf{Z}$ intercepts all causal paths from $\mathbf{X}$ to $\mathbf{Y}$.

1: **function** GETCAND2NDFDC $(\mathcal{G}, \mathbf{X}, \mathbf{I}, \mathbf{R})$
2:     **Output:** $\mathbf{R}'$ with $\mathbf{I} \subseteq \mathbf{R}' \subseteq \mathbf{R}$, the set of candidate variables consisting of all the variables $v \in \mathbf{R}$ such that there is no BD path from $\mathbf{X}$ to $v$.
3:     $\mathbf{R}' \leftarrow \mathbf{R}$
4:     **for all** $v \in \mathbf{R}$:
5:         **if** TESTSEP $(\mathcal{G}_{\underline{\mathbf{X}}}, \mathbf{X}, v, \emptyset) =$ False **then:**
6:             **if** $v \in \mathbf{I}$ **then: return** $\perp$
7:             **else:** $\mathbf{R}' \leftarrow \mathbf{R}' \backslash \{v\}$
8:     **end for**
9:     **return** $\mathbf{R}'$
10: **end function**

Figure 2: A function that outputs the set of candidate variables satisfying the second condition of the FD criterion.

## Step 1 of FINDFDSET

In Step 1, FINDFDSET calls the function GETCAND2NDFDC (presented in Fig. 2) to construct a set $\mathbf{R}'$ that consists of all the variables $v \in \mathbf{R}$ such that there is no BD path from $\mathbf{X}$ to $v$ ($\mathbf{R}'$ is set to empty if there is a BD path from $\mathbf{X}$ to $\mathbf{I}$). Then, there is no BD path from $\mathbf{X}$ to any set $\mathbf{I} \subseteq \mathbf{Z} \subseteq \mathbf{R}'$ since, by definition, there is no BD path from $\mathbf{X}$ to $\mathbf{Z}$ if and only if there is no BD path from $\mathbf{X}$ to any $v \in \mathbf{Z}$.

GETCAND2NDFDC iterates through each variable $v \in \mathbf{R}$ and checks if there exists an open BD path from $\mathbf{X}$ to $v$ by calling the function TESTSEP$(\mathcal{G}_{\underline{\mathbf{X}}}, \mathbf{X}, v, \emptyset)$ [41]. TESTSEP$(\mathcal{G}, \mathbf{A}, \mathbf{B}, \mathbf{C})$ returns True if $\mathbf{C}$ is a separator of $\mathbf{A}$ and $\mathbf{B}$ in $\mathcal{G}$, or False otherwise. Therefore, TESTSEP$(\mathcal{G}_{\underline{\mathbf{X}}}, \mathbf{X}, v, \emptyset)$ returns True if $\emptyset$ is a separator of $\mathbf{X}$ and $v$ in $\mathcal{G}_{\underline{\mathbf{X}}}$ (i.e., there is no BD path from $\mathbf{X}$ to $v$), or False otherwise. If TESTSEP returns False, then $v$ is removed from $\mathbf{R}'$ because every set $\mathbf{Z}$ containing $v$ violates the second condition of the FD criterion relative to $(\mathbf{X}, \mathbf{Y})$.

**Example 2.** Continuing Example 1. With $\mathbf{I} = \emptyset$ and $\mathbf{R} = \{A, B, C, D\}$, GETCAND2NDFDC outputs a set $\mathbf{R}' = \{A, B, C\}$. $D$ is excluded from $\mathbf{R}'$ since there exists a BD path from $\{X\}$ to $\{D\}$, and any set containing $D$ violates the second condition of the FD criterion relative to $(\{X\}, \{Y\})$.

**Lemma 1** (Correctness of GETCAND2NDFDC). GETCAND2NDFDC$(\mathcal{G}, \mathbf{X}, \mathbf{I}, \mathbf{R})$ *generates a set of variables $\mathbf{R}'$ with $\mathbf{I} \subseteq \mathbf{R}' \subseteq \mathbf{R}$ such that $\mathbf{R}'$ consists of all and only variables $v$ that satisfies the second condition of the FD criterion relative to $(\mathbf{X}, \mathbf{Y})$. Further, every subset $\mathbf{Z} \subseteq \mathbf{R}'$ satisfies the second condition of the FD criterion relative to $(\mathbf{X}, \mathbf{Y})$, and every set $\mathbf{Z}$ with $\mathbf{I} \subseteq \mathbf{Z} \subseteq \mathbf{R}$ that satisfies the second condition of the FD criterion relative to $(\mathbf{X}, \mathbf{Y})$ must be a subset of $\mathbf{R}'$.*

**Step 2 of FINDFDSET**

In Step 2, FINDFDSET calls the function GETCAND3RDFDC presented in Fig. 3 to generate a set $\mathbf{R}''$ consisting of all the variables $v \in \mathbf{R}'$ such that there exists a set $\mathbf{Z}$ containing $v$ with $\mathbf{I} \subseteq \mathbf{Z} \subseteq \mathbf{R}'$ that further satisfies the third condition of the FD criterion relative to $(\mathbf{X}, \mathbf{Y})$ (i.e., all BD paths from $\mathbf{Z}$ to $\mathbf{Y}$ are blocked by $\mathbf{X}$). In other words, $\mathbf{R}''$ is the union of all $\mathbf{Z}$ with $\mathbf{I} \subseteq \mathbf{Z} \subseteq \mathbf{R}'$ that satisfies the third condition of the FD criterion.

GETCAND3RDFDC iterates through each variable $v \in \mathbf{R}'$ and calls the function GETDEP$(\mathcal{G}, \mathbf{X}, \mathbf{Y}, \{v\}, \mathbf{R}')$ in line 5. Presented in Fig. 4, GETDEP returns a subset $\mathbf{Z}' \subseteq \mathbf{R}' \setminus \{v\}$ such that all BD paths from $\mathbf{Z} = \{v\} \cup \mathbf{Z}'$ to $\mathbf{Y}$ are blocked by $\mathbf{X}$ (if there exists such $\mathbf{Z}'$). If GETDEP returns $\perp$, then there exists no $\mathbf{Z}$ containing $v$ that satisfies the third condition of the FD criterion relative to $(\mathbf{X}, \mathbf{Y})$, so $v$ is removed from $\mathbf{R}''$.

```
1: function GETCAND3RDFDC(𝒢, X, Y, I, R′)
2:     Output: R″ consisting of all the variables
         v ∈ R′ such that there exists a set Z containing v
         with I ⊆ Z ⊆ R′ that satisfies the third condition
         of the FD criterion relative to (X, Y).
3:     R″ ← R′
4:     for all v ∈ R′:
5:         if GETDEP(𝒢, X, Y, {v}, R′) = ⊥ then:
6:             if v ∈ I then: return ⊥
7:             else: R″ ← R″ \ {v}
8:     end for
9:     return R″
10: end function
```

Figure 3: A function that outputs the set of candidate variables potentially satisfying the second and third conditions of the FD criterion.

**Example 3.** Continuing Example 2. Given $\mathbf{I} = \emptyset$ and $\mathbf{R}' = \{A, B, C\}$, GETCAND3RDFDC outputs $\mathbf{R}'' = \{A, B, C\}$ because for each variable $v \in \mathbf{R}''$, GETDEP finds a set $\mathbf{Z}'$ such that $\{v\} \cup \mathbf{Z}'$ satisfies the third condition of the FD criterion relative to $(\{X\}, \{Y\})$. For $v = A$, $\mathbf{Z}' = \emptyset$, for $v = B$, $\mathbf{Z}' = \{A\}$, and for $v = C$, $\mathbf{Z}' = \{A\}$.

Next, we explain how the function GETDEP$(\mathcal{G}, \mathbf{X}, \mathbf{Y}, \mathbf{T}, \mathbf{R}')$ works. First, GETDEP constructs an undirected graph $\mathcal{M}$ in a way that the paths from $\mathbf{T}$ to $\mathbf{Y}$ in $\mathcal{M}$ represent all BD paths from $\mathbf{T}$ to $\mathbf{Y}$ that cannot be blocked by $\mathbf{X}$ in $\mathcal{G}$. The auxiliary function MORALIZE$(\mathcal{G})$ moralizes a given graph $\mathcal{G}$ into an undirected graph. The moralization is performed on the subgraph over $An(\mathbf{T} \cup \mathbf{X} \cup \mathbf{Y})$ instead of $\mathcal{G}$ based on the following property: $\mathbf{T}$ and $\mathbf{Y}$ are $d$-separated by $\mathbf{X}$ in $\mathcal{G}$ if and only if $\mathbf{X}$ is a $\mathbf{T}$-$\mathbf{Y}$ node cut (i.e., removing $\mathbf{X}$ disconnects $\mathbf{T}$ and $\mathbf{Y}$) in $\mathcal{G}' = $ MORALIZE$(\mathcal{G}_{An(\mathbf{T} \cup \mathbf{X} \cup \mathbf{Y})})$ [21].

GETDEP performs Breadth-First Search (BFS) from $\mathbf{T}$ to $\mathbf{Y}$ on $\mathcal{M}$ and incrementally constructs a subset $\mathbf{Z}' \subseteq \mathbf{R}' \setminus \mathbf{T}$ such that, after BFS terminates, there will be no BD path from $\mathbf{Z} = \mathbf{T} \cup \mathbf{Z}'$ to $\mathbf{Y}$ that cannot be blocked by $\mathbf{X}$ in $\mathcal{G}$. While constructing $\mathbf{Z}'$, GETDEP calls the function GETNEIGHBORS$(u, \mathcal{M})$ (presented in Fig. 8, Appendix) to obtain all observed neighbors of $u$ in $\mathcal{M}$.

The BFS starts from each variable $v \in \mathbf{T}$. Whenever a non-visited node $u$ is encountered, the set $\mathbf{NR}$, observed neighbors of $u$ that belong to $\mathbf{R}'$, is computed. $\mathbf{NR}$ can be added to $\mathbf{Z}'$ because removing all outgoing edges of $\mathbf{NR}$ may contribute to disconnecting some BD paths $\Pi$ from $\mathbf{T}$ to $\mathbf{Y}$ that cannot be blocked by $\mathbf{X}$ in $\mathcal{G}$. In other words, in $\mathcal{G}_{\overline{\mathbf{T} \cup \mathbf{Z}' \cup \mathbf{NR}}}$, $\Pi$ could be disconnected from $\mathbf{T}$ to

```
 1: function GETDEP($\mathcal{G}, \mathbf{X}, \mathbf{Y}, \mathbf{T}, \mathbf{R}'$)
 2:     Output: $\mathbf{Z}' \subseteq \mathbf{R}' \setminus \mathbf{T}$, a set of variables such that $\mathbf{T} \cup \mathbf{Z}'$ satisfies the third condition of the
        FD criterion relative to $(\mathbf{X}, \mathbf{Y})$.
 3:     $\mathcal{G}' \leftarrow \mathcal{G}_{An(\mathbf{T} \cup \mathbf{X} \cup \mathbf{Y})}$
 4:     $\mathcal{G}' \leftarrow \mathcal{G}'$ with all bidirected edges $A \leftrightarrow B$ replaced by a latent node $L_{AB}$ and two edges
        $L_{AB} \to A$ and $L_{AB} \to B$
 5:     $\mathcal{G}'' \leftarrow \mathcal{G}'_{\underline{\mathbf{T}}}$
 6:     $\mathcal{M} \leftarrow$ MORALIZE($\mathcal{G}''$) then remove $\mathbf{X}$
 7:     $\mathbf{Z}' \leftarrow \emptyset, \mathbf{Q} \leftarrow \mathbf{T}$ and mark all $v \in \mathbf{T}$ as visited
 8:     while $\mathbf{Q} \neq \emptyset$ do
 9:         $u \leftarrow \mathbf{Q}$.POP()
10:         if $u \in \mathbf{Y}$ then: return $\perp$
11:         $\mathbf{NR} \leftarrow$ GETNEIGHBORS($u, \mathcal{M}$) $\cap \mathbf{R}'$ that are not visited
12:         $\mathcal{G}'' \leftarrow \mathcal{G}'_{\underline{\mathbf{T} \cup \mathbf{Z}' \cup \mathbf{NR}}}$
13:         $\mathcal{M} \leftarrow$ MORALIZE($\mathcal{G}''$) then remove $\mathbf{X}$
14:         $\mathbf{N}' \leftarrow$ GETNEIGHBORS($u, \mathcal{M}$) that are not visited
15:         $\mathbf{NR}' \leftarrow \{w \in \mathbf{NR}|$ there exists an incoming arrow into $w$ in $\mathcal{G}\}$
16:         $\mathbf{N} \leftarrow \mathbf{N}' \cup \mathbf{NR}', \mathbf{Z}' \leftarrow \mathbf{Z}' \cup \mathbf{NR}$
17:         $\mathbf{Q}$.INSERT($\mathbf{N}$) and mark all $w \in \mathbf{N}$ as visited
18:     end while
19:     return $\mathbf{Z}'$
20: end function
```

Figure 4: A function that facilitates the construction of a set that satisfies the third condition of the FD criterion.

$\mathbf{Y}$ where $\Pi$ are not disconnected in $\mathcal{G}_{\underline{\mathbf{T} \cup \mathbf{Z}'}}$. After adding $\mathbf{NR}$ to $\mathbf{Z}'$, $\mathcal{M}$ must be reconstructed in a way that reflects the setting where all outgoing edges of $\mathbf{NR}$ are removed. BFS will be performed on such modified $\mathcal{M}$.

GETDEP checks if there exists any set of nodes $\mathbf{N}$ to be visited further. $\mathbf{N}$ consists of two sets: 1) $\mathbf{N}'$, all observed neighbors of $u$ that are still reachable from $u$, even after removing all outgoing edges of $\mathbf{NR}$, and 2) $\mathbf{NR}' \subseteq \mathbf{NR}$ where for every node $w \in \mathbf{NR}$, there exists an incoming arrow into $w$ in $\mathcal{G}$. All nodes in $\mathbf{NR}'$ must be checked because there might exist some BD path $\pi$ from $w$ to $y \in \mathbf{Y}$ that cannot be blocked by $\mathbf{X}$ in $\mathcal{G}$. If $\pi$ cannot be disconnected from $w$ to $y$, then the set $\mathbf{Z}$ will violate the third condition of the FD criterion relative to $(\mathbf{X}, \mathbf{Y})$.

The BFS continues until either a node $y \in \mathbf{Y}$ is visited, or no more nodes can be visited. If GETDEP returns a set $\mathbf{Z}'$, then we have that all BD paths from $\mathbf{T}$ to $\mathbf{Y}$ that cannot be blocked by $\mathbf{X}$ in $\mathcal{G}$ have been disconnected in $\mathcal{G}_{\underline{\mathbf{Z}}}$ while ensuring that there exists no BD path from $\mathbf{Z}$ to $\mathbf{Y}$ that cannot be blocked by $\mathbf{X}$ in $\mathcal{G}$. Therefore, $\mathbf{Z}$ satisfies the third condition of the FD criterion relative to $(\mathbf{X}, \mathbf{Y})$. Otherwise, if GETDEP returns $\perp$ (i.e., $y$ is visited), then there does not exist any $\mathbf{Z}$ containing $\mathbf{T}$ that satisfies the third condition of the FD criterion relative to $(\mathbf{X}, \mathbf{Y})$. This is because there exists a BD path $\pi$ from $t \in \mathbf{T}$ to $y$ that cannot be blocked by $\mathbf{X}$ in $\mathcal{G}$; removing outgoing edges of all $w \in \mathbf{R}'$ that intersect $\pi$ cannot disconnect $\pi$ from $t$ to $y$.

**Example 4.** Expanding on Example 3 to show the use of function GETDEP. Consider the case when $v = B$. Then, $\mathbf{Q} = \mathbf{T} = \{B\}$ and $u = B$ is popped from $\mathbf{Q}$ at line 9. We have $\mathbf{NR} = \{A\}, \mathbf{N}' = \emptyset, \mathbf{NR}' = \{A\}, \mathbf{N} = \{A\}$, and $\mathbf{Z}' = \{A\}$. Since $\mathbf{N}$ is inserted to $\mathbf{Q}$ at line 17, $u = A$ is popped from $\mathbf{Q}$ in the next iteration of while loop. Then, $\mathbf{NR} = \emptyset, \mathbf{N}' = \emptyset, \mathbf{NR}' = \emptyset$, and $\mathbf{N} = \emptyset$. Since $\mathbf{Q}$ is empty, the while loop terminates and GETDEP returns $\mathbf{Z}' = \{A\}$.

**Example 5.** Illustrating the use of function GETDEP. Let $\mathbf{I} = \emptyset, \mathbf{R}' = \{B, C\}$, and $v = B$. $\mathbf{Q} = \mathbf{T} = \{B\}$ and $u = B$ is popped from $\mathbf{Q}$ at line 9. $\mathbf{NR} = \emptyset, \mathbf{N}' = \{A\}, \mathbf{NR}' = \emptyset, \mathbf{N} = \{A\}$, and $\mathbf{Z}' = \emptyset$. Since $\mathbf{N}$ is inserted to $\mathbf{Q}$ at line 17, $u = A$ is popped from $\mathbf{Q}$ in the second iteration of while loop. $\mathbf{NR} = \mathbf{NR}' = \{C\}, \mathbf{N}' = \mathbf{N} = \{C, D, Y\}, \mathbf{Z}' = \{C\}$, and $\mathbf{Q} = \{C, D, Y\}$. On the third iteration, $u = C$ is popped from $\mathbf{Q}$. $\mathbf{NR} = \mathbf{NR}' = \mathbf{N}' = \mathbf{N} = \emptyset$ and $\mathbf{Q} = \{D, Y\}$. On the fourth iteration, $u = D$ is popped from $\mathbf{Q}$. $\mathbf{NR} = \mathbf{NR}' = \mathbf{N}' = \mathbf{N} = \emptyset$ and $\mathbf{Q} = \{Y\}$. Next, $u = Y$ is popped from $\mathbf{Q}$. Since $u \in \{Y\}$, GETDEP returns $\perp$ at line 10. There exists no set $\mathbf{Z}' \subseteq (\mathbf{R}' \setminus \mathbf{T}) = \{C\}$ such that $\mathbf{T} \cup \mathbf{Z}'$ satisfies the third condition of the FD criterion relative to $(\{X\}, \{Y\})$.

**Lemma 2** (Correctness of GETCAND3RDFDC). *GETCAND3RDFDC$(\mathcal{G}, \mathbf{X}, \mathbf{Y}, \mathbf{I}, \mathbf{R}')$ in Step 2 of Alg. 1 generates a set of variables $\mathbf{R}''$ where $\mathbf{I} \subseteq \mathbf{R}'' \subseteq \mathbf{R}'$. $\mathbf{R}''$ consists of all and only variables $v$ such that there exists a subset $\mathbf{Z}$ with $\mathbf{I} \subseteq \mathbf{Z} \subseteq \mathbf{R}'$ and $v \in \mathbf{Z}$ that satisfies the third condition of the FD criterion relative to $(\mathbf{X}, \mathbf{Y})$. Further, every set $\mathbf{Z}$ with $\mathbf{I} \subseteq \mathbf{Z} \subseteq \mathbf{R}$ that satisfies both the second and the third conditions of the FD criterion must be a subset of $\mathbf{R}''$.*

**Remark:** Even though every set $\mathbf{Z}$ with $\mathbf{I} \subseteq \mathbf{Z} \subseteq \mathbf{R}'$ that satisfies the third condition of the FD criterion must be a subset of $\mathbf{R}''$, *not* every subset $\mathbf{Z} \subseteq \mathbf{R}''$ satisfies the third condition of the FD criterion, as illustrated by the following example.

**Example 6.** In Example 3, GETCAND3RDFDC outputs $\mathbf{R}'' = \{A, B, C\}$. However, for $\mathbf{Z} = \{B\}$, the BD path $\{B \leftarrow A \rightarrow D \rightarrow Y\}$ is not blocked by $\{X\}$; for $\mathbf{Z} = \{C\}$, the BD path $\{C \leftarrow A \rightarrow D \rightarrow Y\}$ is not blocked by $\{X\}$.

On the other hand, we show that $\mathbf{Z} = \mathbf{R}''$ itself satisfies the third condition of the FD criterion, as shown in the following.

**Lemma 3.** $\mathbf{R}''$ *generated by* GETCAND3RDFDC *(in Step 2 of Alg. 1) satisfies the third condition of the FD criterion, that is, all BD paths from $\mathbf{R}''$ to $\mathbf{Y}$ are blocked by $\mathbf{X}$.*

## Step 3 of FINDFDSET

Finally, in Step 3, FINDFDSET looks for a set $\mathbf{Z} \subseteq \mathbf{R}''$ that satisfies the first condition of the FD criterion relative to $(\mathbf{X}, \mathbf{Y})$, that is, $\mathbf{Z}$ intercepts all causal paths from $\mathbf{X}$ to $\mathbf{Y}$. To facilitate checking whether a set $\mathbf{Z}$ intercepts all causal paths from $\mathbf{X}$ to $\mathbf{Y}$, we introduce the concept of causal path graph defined as follows.

**Definition 2.** (Causal Path Graph) Let $\mathcal{G}$ be a causal graph and $\mathbf{X}, \mathbf{Y}$ disjoint sets of variables. A *causal path graph* $\mathcal{G}'$ relative to $(\mathcal{G}, \mathbf{X}, \mathbf{Y})$ is a graph over $\mathbf{X} \cup \mathbf{Y} \cup PCP(\mathbf{X}, \mathbf{Y})$, where $PCP(\mathbf{X}, \mathbf{Y}) = (De(\mathbf{X})_{\mathcal{G}_{\overline{\mathbf{X}}}} \setminus \mathbf{X}) \cap An(\mathbf{Y})_{\mathcal{G}_{\underline{\mathbf{X}}}}{}^2$, constructed as follows:

1. Construct a subgraph $\mathcal{G}'' = \mathcal{G}_{\mathbf{X} \cup \mathbf{Y} \cup PCP(\mathbf{X}, \mathbf{Y})}$.

2. Construct a graph $\mathcal{G}' = \mathcal{G}''_{\overline{\mathbf{X}}\underline{\mathbf{Y}}}$, then remove all bidirected edges from $\mathcal{G}'$.

A function GETCAUSALPATHGRAPH$(\mathcal{G}, \mathbf{X}, \mathbf{Y})$ for constructing a causal path graph is presented in Fig. 9 in the Appendix.

**Example 7.** Consider the causal graph $\mathcal{G}'$ shown in Fig. 1b with $\mathbf{X} = \{X\}$, and $\mathbf{Y} = \{Y\}$. The causal path graph $\mathcal{G}''$ relative to $(\mathcal{G}', \{X\}, \{Y\})$ is shown in Fig. 5b. All causal paths from $\{X\}$ to $\{Y\}$ in $\mathcal{G}'$ are present in $\mathcal{G}''$.

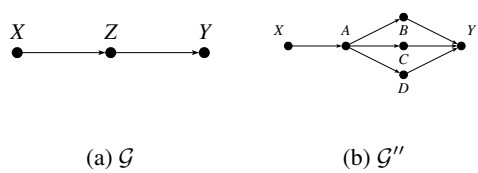

(a) $\mathcal{G}$          (b) $\mathcal{G}''$

Figure 5: Two causal path graphs generated from (a) the causal graph in Fig. 1a, and (b) the causal graph in Fig. 1b. Both preserve all and only causal paths from $\{X\}$ to $\{Y\}$ in the original graphs.

After constructing a causal path graph $\mathcal{G}'$ relative to $(\mathcal{G}, \mathbf{X}, \mathbf{Y})$, we use the function TESTSEP$(\mathcal{G}', \mathbf{X}, \mathbf{Y}, \mathbf{Z})$ to check if $\mathbf{Z}$ is a separator of $\mathbf{X}$ and $\mathbf{Y}$ in $\mathcal{G}'$. Based on the following lemma, $\mathbf{Z}$ satisfies the first condition of the FD criterion relative to $(\mathbf{X}, \mathbf{Y})$ if and only if TESTSEP returns True.

**Lemma 4.** *Let $\mathcal{G}$ be a causal graph and $\mathbf{X}, \mathbf{Y}, \mathbf{Z}$ disjoint sets of variables. Let $\mathcal{G}'$ be the causal path graph relative to $(\mathcal{G}, \mathbf{X}, \mathbf{Y})$. Then, $\mathbf{Z}$ satisfies the first condition of the FD criterion relative to $(\mathbf{X}, \mathbf{Y})$ if and only if $\mathbf{Z}$ is a separator of $\mathbf{X}$ and $\mathbf{Y}$ in $\mathcal{G}'$.*

Given the set $\mathbf{R}''$ that contains every set $\mathbf{Z}$ with $\mathbf{I} \subseteq \mathbf{Z} \subseteq \mathbf{R}$ that satisfies both the second and the third conditions of the FD criterion (Lemma 2), it may appear that we need to search for a set $\mathbf{Z} \subseteq \mathbf{R}''$ that satisfies the first condition of the FD criterion. We show instead that all we need is to check whether the set $\mathbf{R}''$ itself satisfies the first condition which has been shown to satisfy the second and third conditions by Lemma 3. This result is summarized in the following lemma.

---

[2]A notation introduced by van der Zander et al. [41] to denote the set of variables on proper causal paths from $\mathbf{X}$ to $\mathbf{Y}$.

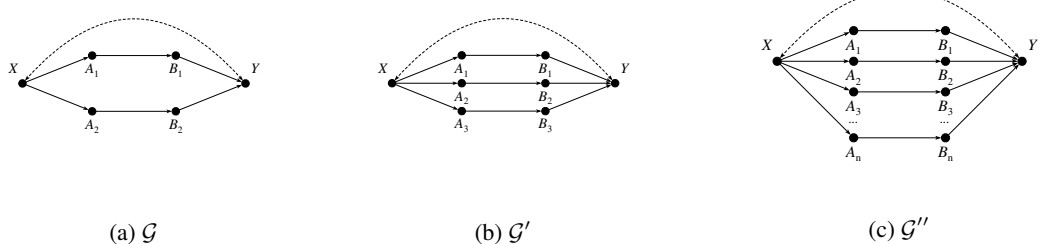

$$(a)\ \mathcal{G} \qquad\qquad (b)\ \mathcal{G}' \qquad\qquad (c)\ \mathcal{G}''$$

Figure 6: Three examples of the FD criterion to demonstrate that total number of FD adjustment sets may be exponential with respect to the number of nodes in a graph.

**Lemma 5.** *There exists a set $\mathbf{Z}_0$ satisfying the FD criterion relative to $(\mathbf{X}, \mathbf{Y})$ with $\mathbf{I} \subseteq \mathbf{Z}_0 \subseteq \mathbf{R}$ if and only if $\mathbf{R}''$ generated by* GETCAND3RDFDC *(in Step 2 of Alg. 1) satisfies the FD criterion relative to $(\mathbf{X}, \mathbf{Y})$.*

**Example 8.** Continuing Example 3. In Step 3, FINDFDSET outputs $\mathbf{Z} = \mathbf{R}'' = \{A, B, C\}$ since $\mathbf{Z}$ is a separator of $\{X\}$ and $\{Y\}$ in the causal path graph $\mathcal{G}''$ in Fig. 5b.

The results in this section are summarized as follows.

**Theorem 1** (Correctness of FINDFDSET)**.** *Let $\mathcal{G}$ be a causal graph, $\mathbf{X}, \mathbf{Y}$ disjoint sets of variables, and $\mathbf{I}, \mathbf{R}$ sets of variables such that $\mathbf{I} \subseteq \mathbf{R}$. Then,* FINDFDSET$(\mathcal{G}, \mathbf{X}, \mathbf{Y}, \mathbf{I}, \mathbf{R})$ *outputs a set $\mathbf{Z}$ with $\mathbf{I} \subseteq \mathbf{Z} \subseteq \mathbf{R}$ that satisfies the FD criterion relative to $(\mathbf{X}, \mathbf{Y})$, or outputs $\perp$ if none exists, in $O(n^3(n+m))$ time, where $n$ and $m$ represent the number of nodes and edges in $\mathcal{G}$.*

## 4    Enumerating Front-door Adjustment Sets

Our goal in this section is to develop an algorithm that lists *all* FD adjustment sets in a causal diagram. In general, there may exist exponential number of such sets, which means that any listing algorithm will take exponential time to list them all. We will instead look for an algorithm that has an interesting property known as *polynomial delay* [38]. In words, poly-delay algorithms output the first answer (or indicate none is available) in polynomial time, and take polynomial time to output each consecutive answer as well. Consider the following example.

**Example 9.** Consider the three causal graphs in Fig. 6. In $\mathcal{G}$ shown in Fig. 6a, there exists 9 valid FD adjustment sets relative to

---

**Algorithm 2** LISTFDSETS $(\mathcal{G}, \mathbf{X}, \mathbf{Y}, \mathbf{I}, \mathbf{R})$

1: **Input:** $\mathcal{G}$ a causal diagram; $\mathbf{X}, \mathbf{Y}$ disjoint sets of variables; $\mathbf{I}, \mathbf{R}$ sets of variables.
2: **Output:** Listing front-door adjustment set $\mathbf{Z}$ relative to $(\mathbf{X}, \mathbf{Y})$ where $\mathbf{I} \subseteq \mathbf{Z} \subseteq \mathbf{R}$.
3: **if** FINDFDSET$(\mathcal{G}, \mathbf{X}, \mathbf{Y}, \mathbf{I}, \mathbf{R}) \neq \perp$ **then:**
4:     **if** $\mathbf{I} = \mathbf{R}$ **then:** Output $\mathbf{I}$
5:     **else:**
6:         $v \leftarrow$ any variable from $\mathbf{R} \setminus \mathbf{I}$
7:         LISTFDSETS$(\mathcal{G}, \mathbf{X}, \mathbf{Y}, \mathbf{I} \cup \{v\}, \mathbf{R})$
8:         LISTFDSETS$(\mathcal{G}, \mathbf{X}, \mathbf{Y}, \mathbf{I}, \mathbf{R} \setminus \{v\})$

---

$(\{X\}, \{Y\})$. In $\mathcal{G}'$, presented in Fig. 6b, two variables $A_3$ and $B_3$ are added from $\mathcal{G}$, forming an additional causal path from $X$ to $Y$. 27 FD adjustment sets relative to $(\{X\}, \{Y\})$ are available in $\mathcal{G}'$. If another causal path $X \to A_4 \to B_4 \to Y$ is added to $\mathcal{G}'$, then there are 81 FD adjustment sets relative to $(\{X\}, \{Y\})$. As shown in Fig. 6c, in a graph $\mathcal{G}''$ with similar pattern with causal path $X \to A_i \to B_i \to Y, i = 1, \ldots n$, there are at least $3^n$ number of FD adjustment sets.

We have developed an algorithm named LISTFDSETS, shown in Alg. 2, that lists all FD adjustment sets $\mathbf{Z}$ relative to $(\mathbf{X}, \mathbf{Y})$ satisfying $\mathbf{I} \subseteq \mathbf{Z} \subseteq \mathbf{R}$ with polynomial delay, given a causal diagram $\mathcal{G}$, disjoint sets of variables $\mathbf{X}$ and $\mathbf{Y}$, and two sets of variables $\mathbf{I}$ and $\mathbf{R}$.

**Example 10.** Consider the causal graph $\mathcal{G}'$ shown in Fig. 1b with $\mathbf{X} = \{X\}, \mathbf{Y} = \{Y\}, \mathbf{I} = \emptyset$ and $\mathbf{R} = \{A, B, C, D\}$. LISTFDSETS outputs $\{A, B, C\}, \{A, B\}, \{A, C\}, \{A\}$ one by one, and finally stops as no more adjustment sets exist.

The algorithm LISTFDSETS takes the same search strategy as the listing algorithm LISTSEP [41] that enumerates all BD adjustment sets with polynomial delay. LISTFDSETS implicitly constructs a binary search tree where each tree node $\mathcal{N}(\mathbf{I}', \mathbf{R}')$ represents the collection of all FD adjustment sets $\mathbf{Z}$ relative to $(\mathbf{X}, \mathbf{Y})$ with $\mathbf{I}' \subseteq \mathbf{Z} \subseteq \mathbf{R}'$. The search starts from the root tree node $\mathcal{N}(\mathbf{I}, \mathbf{R})$, indicating that LISTFDSETS will list all FD adjustment sets $\mathbf{Z}$ relative to $(\mathbf{X}, \mathbf{Y})$ with $\mathbf{I} \subseteq \mathbf{Z} \subseteq \mathbf{R}$.

Upon visiting a node $\mathcal{N}(\mathbf{I}', \mathbf{R}')$, LISTFDSETS first calls the function FINDFDSET (line 3) to decide whether it is necessary to search further from $\mathcal{N}$. If FINDFDSET outputs $\bot$, then there does not exist any FD adjustment set $\mathbf{Z}_0$ with $\mathbf{I}' \subseteq \mathbf{Z}_0 \subseteq \mathbf{R}'$ and there is no need to search further. Otherwise, $\mathcal{N}$ spawns two children, $\mathcal{N}_1$ and $\mathcal{N}_2$, and LISTFDSETS continues the search over each child separately. $\mathcal{N}_1$ in line 7 represents the collection of all FD adjustment sets $\mathbf{Z}_1$ relative to $(\mathbf{X}, \mathbf{Y})$ where $\mathbf{I}' \cup \{v\} \subseteq \mathbf{Z}_1 \subseteq \mathbf{R}'$. On the other hand, $\mathcal{N}_2$ in line 8 represents the collection of all FD adjustment sets $\mathbf{Z}_2$ where $\mathbf{I}' \subseteq \mathbf{Z}_2 \subseteq \mathbf{R}' \setminus \{v\}$. $\mathcal{N}_1$ and $\mathcal{N}_2$ are disjoint

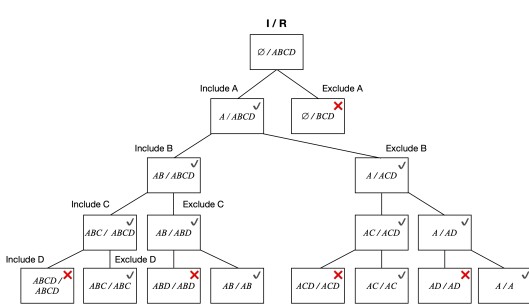

Figure 7: A search tree illustrating the running of LISTFDSETS in Example 11.

and thus the search never overlaps, which is crucial to guaranteeing that LISTFDSETS runs in polynomial delay. Finally, a leaf tree node $\mathcal{L}$ is reached when $\mathbf{I}' = \mathbf{R}'$, and LISTFDSETS outputs a valid FD adjustment set $\mathbf{I}'$.

**Example 11.** Continuing from Example 10. Fig. 7 shows a search tree generated by running LISTFDSETS($\mathcal{G}', \{X\}, \{Y\}, \emptyset, \{A, B, C, D\}$). Initially, the search starts from the root tree node $\mathcal{N}(\emptyset, \{A, B, C, D\})$. Since FINDFDSET returns a set $\{A, B, C\}$, $\mathcal{N}$ branches out into two children $\mathcal{N}'(\{A\}, \{A, B, C, D\})$ and $\mathcal{N}''(\emptyset, \{B, C, D\})$. The search continues from the left child $\mathcal{N}'$ until reaching the leaf tree node $\mathcal{L}_1(\{A, B, C, D\}, \{A, B, C, D\})$ where FINDFDSET returns $\bot$. LISTFD-SETS backtracks to the parent tree node $\mathcal{N}_1(\{A, B, C\}, \{A, B, C, D\})$ and then checks the next leaf $\mathcal{L}_2(\{A, B, C\}, \{A, B, C\})$ where FINDFDSET returns a set $\{A, B, C\}$, a valid FD admissible set relative to $(\{X\}, \{Y\})$. LISTFDSETS outputs $\{A, B, C\}$. Next, LISTFDSETS backtracks to the tree node $\mathcal{N}_2(\{A, B\}, \{A, B, C, D\})$ and reaches the leaf $\mathcal{L}_3(\{A, B\}, \{A, B\})$ where FINDFDSET outputs $\{A, B\}$, and thus LISTFDSETS outputs $\{A, B\}$. LISTFDSETS continues and outputs two sets $\{A, C\}$ and $\{A\}$ in order. Finally, LISTFDSETS backtracks to the root $\mathcal{N}$ and checks the right child $\mathcal{N}''$ where FINDFDSET returns $\bot$. LISTFDSETS does not search further from $\mathcal{N}''$ and stops as no more tree node is left to be visited.

Our results are summarized in the following theorem, which provides the correctness, completeness, and poly-delay complexity of the proposed algorithm. Note that the completeness of the algorithm means that it lists "all" valid sets satisfying the FD criterion. On the other hand, Pearl's FD criterion is not complete in the sense that there might exist a causal effect that can be computed by the FD adjustment formula (Eq. (3)) but the set $\mathbf{Z}$ does not satisfy the FD criterion.

**Theorem 2** (Correctness of LISTFDSETS). *Let $\mathcal{G}$ be a causal graph, $\mathbf{X}, \mathbf{Y}$ disjoint sets of variables, and $\mathbf{I}, \mathbf{R}$ sets of variables. LISTFDSETS($\mathcal{G}, \mathbf{X}, \mathbf{Y}, \mathbf{I}, \mathbf{R}$) enumerates all and only sets $\mathbf{Z}$ with $\mathbf{I} \subseteq \mathbf{Z} \subseteq \mathbf{R}$ that satisfy the FD criterion relative to $(\mathbf{X}, \mathbf{Y})$ in $O(n^4(n + m))$ delay where $n$ and $m$ represent the number of nodes and edges in $\mathcal{G}$.*

## 5 Discussion and Conclusions

This work has some limitations and can be extended in several directions. First, Pearl's FD criterion is not complete with respect to the FD adjustment formula (Eq. (3)). While the BD criterion has been generalized to a complete criterion for BD adjustment [35], it is an interesting open problem to come up with a complete criterion for sets satisfying the FD adjustment. Second, this work assumes that the causal diagram is given (or inferred based on scientists' domain knowledge and/or data). Although this assumption is quite common throughout the causal inference literature, more recent work has moved to finding BD admissible sets given incomplete or partially specified causal diagrams, e.g., maximal ancestral graphs (MAGs) [41], partial ancestral graphs (PAGs) [29], and completed partially directed acyclic graphs (CPDAGs) [29]. There are algorithms capable of performing causal effect identification in a data-driven fashion from an equivalence class [14, 15, 16, 17]. It is an

interesting and certainly challenging future work to develop algorithms for finding FD admissible sets in these types of graphs. Some recent work has proposed data-driven methods for finding and listing BD admissible sets, using an anchor variable, when the underlying causal diagram is unknown [7, 6, 33]. A criterion for testing FD-admissibility of a given set using data and an anchor variable is also available [4]. Other interesting future research topics include developing algorithms for finding minimal, minimum, and minimum cost FD adjustment sets, which are available for the BD adjustment sets [42], as well as algorithms for finding conditional FD adjustment sets [13, 9]. Having said all of that, we believe that the results developed in this paper is a necessary step towards solving these more challenging problems.

After all, we started from the observation that identification is not restricted to BD adjustment, and Pearl's FD criterion provides a classic strategy for estimating causal effects from observational data and qualitative knowledge encoded in the form of a causal diagram. The criterion is drawing more attention in recent years and statistically efficient and doubly robust estimators have been developed for estimating the FD estimand from finite samples. In this paper, we develop algorithms that given a causal diagram $\mathcal{G}$, find an admissible FD set (Alg. 1 FINDFDSET, Thm. 1) and enumerate all admissible FD sets with polynomial delay (Alg. 2 LISTFDSETS, Thm. 2). We hope that the methods and algorithms proposed in this work will help scientists to use the FD strategy for causal effects estimation in the practical applications and are useful for scientists in study design to select covariates based on desired properties, including cost, feasibility, and statistical power.

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
