# A Appendix

**Lemma 1** (Correctness of GETCAND2NDFDC). GETCAND2NDFDC$(\mathcal{G}, \mathbf{X}, \mathbf{I}, \mathbf{R})$ *generates a set of variables* $\mathbf{R}'$ *with* $\mathbf{I} \subseteq \mathbf{R}' \subseteq \mathbf{R}$ *such that* $\mathbf{R}'$ *consists of all and only variables* $v$ *that satisfies the second condition of the FD criterion relative to* $(\mathbf{X}, \mathbf{Y})$. *Further, every subset* $\mathbf{Z} \subseteq \mathbf{R}'$ *satisfies the second condition of the FD criterion relative to* $(\mathbf{X}, \mathbf{Y})$, *and every set* $\mathbf{Z}$ *with* $\mathbf{I} \subseteq \mathbf{Z} \subseteq \mathbf{R}$ *that satisfies the second condition of the FD criterion relative to* $(\mathbf{X}, \mathbf{Y})$ *must be a subset of* $\mathbf{R}'$.

*Proof.* GETCAND2NDFDC iterates through every node $v \in \mathbf{R}$. For each $v$, the function TESTSEP$(\mathcal{G}_{\underline{\mathbf{X}}}, \mathbf{X}, v, \emptyset)$ is called in line 5 to check if $\emptyset$ is a separator of $\mathbf{X}$ and $v$ in $\mathcal{G}_{\underline{\mathbf{X}}}$, i.e., whether there exists an open BD path from $\mathbf{X}$ to $v$ or not. If TESTSEP returns True, then there is no open BD path from $\mathbf{X}$ to $v$ and $v$ satisfies the second condition of the FD criterion relative to $(\mathbf{X}, \mathbf{Y})$. In this case, $v$ is kept in $\mathbf{R}'$. Otherwise, if TESTSEP returns False, then there exists an open BD path from $\mathbf{X}$ to $v$. By definition, for every set $\mathbf{Z}$ that includes $v$, there exists an open BD path from $\mathbf{X}$ to $\mathbf{Z}$. $\mathbf{Z}$ violates the second condition of the FD criterion relative to $(\mathbf{X}, \mathbf{Y})$, and thus $v$ is removed from $\mathbf{R}'$. A special case is when $v \in \mathbf{I}$. GETCAND2NDFDC returns $\bot$ because $\mathbf{R}'$ will not include any subset $\mathbf{Z}$ with $\mathbf{I} \subseteq \mathbf{Z}$ that satisfies the second condition of the FD criterion relative to $(\mathbf{X}, \mathbf{Y})$.

At the end of the function, GETCAND2NDFDC has generated a set $\mathbf{R}'$ that includes all and only variables $v$ that satisfies the second condition of the FD criterion relative to $(\mathbf{X}, \mathbf{Y})$. By definition, there exists no BD path from $\mathbf{X}$ to $\mathbf{Z}$ if and only if there exists no BD path from every $x \in \mathbf{X}$ to every $v \in \mathbf{Z}$. Hence, every subset $\mathbf{Z} \subseteq \mathbf{R}'$ satisfies the second condition of the FD criterion relative to $(\mathbf{X}, \mathbf{Y})$, and $\mathbf{R}'$ contains all and only sets $\mathbf{Z}$ with $\mathbf{I} \subseteq \mathbf{Z} \subseteq \mathbf{R}$ that satisfies the second condition of the FD criterion relative to $(\mathbf{X}, \mathbf{Y})$. $\qquad\square$

**Proposition A.1** (Complexity of GETCAND2NDFDC). GETCAND2NDFDC *runs in* $O(n(n+m))$ *time where* $n$ *and* $m$ *represent the number of nodes and edges in* $\mathcal{G}$.

*Proof.* GETCAND2NDFDC iterates through all variables in $\mathbf{R}$ of size $O(n)$. For each variable $v \in \mathbf{R}$, the function TESTSEP is called, which takes $O(n+m)$ time [42]. $\qquad\square$

**Proposition A.2** (Correctness of GETNEIGHBORS). *Let* $\mathcal{G}$ *be an undirected graph and* $v$ *a variable in* $\mathcal{G}$. GETNEIGHBORS *correctly outputs all observed neighbors* $\mathbf{N}$ *of* $v$ *in* $\mathcal{G}$. GETNEIGHBORS *runs in* $O(n+m)$ *time where* $n$ *and* $m$ *represent the number of nodes and edges in* $\mathcal{G}$.

*Proof.* GETNEIGHBORS computes $\mathbf{N}$, all adjacent nodes of $v$ in $\mathcal{G}$ that are observed. Also, all latent adjacent nodes $\mathbf{L}$ of $v$ need to be considered because there might exist some observed adjacent nodes $\mathbf{O}$ of $\mathbf{L}$ where $\mathbf{O}$ belongs to observed neighbors of $v$. If $\mathbf{L}$ is empty, then all adjacent nodes of $v$ are observed, and thus GETNEIGHBORS returns $\mathbf{N}$. Otherwise, GETNEIGHBORS performs BFS from $\mathbf{L}$, searching for all observed neighbors of $\mathbf{L}$. The nodes $v$, $\mathbf{N}$ and $\mathbf{L}$ are marked as visited to guarantee that the nodes will not be visited more than once.

When BFS is performed, one latent node $u$ is popped from $\mathbf{Q}$ at a time. Then, all observed adjacent nodes $\mathbf{O}$ of $u$ (that have not been visited before) are computed and added to $\mathbf{N}$. Further, there may exist some latent adjacent nodes $\mathbf{L}'$ of $u$ that have not been visited, and then there may exist some observed neighbors of $\mathbf{L}'$ as well. Hence, $\mathbf{L}'$ is inserted into $\mathbf{Q}$ and all nodes in $\mathbf{L}'$ is marked as visited. The procedure continues until $\mathbf{Q}$ becomes empty.

At the end of while loop, $\mathbf{N}$ must include all and only observed neighbors of $v$ in $\mathcal{G}$ because all observed adjacent nodes of $v$ are added to $\mathbf{N}$, and for all latent adjacent nodes $\mathbf{L}$ of $v$, all observed neighbors of $\mathbf{L}$ are also added to $\mathbf{N}$.

GETNEIGHBORS runs in $O(n+m)$ time because, while performing BFS, every node and edge in $\mathcal{G}$ will be visited at most once. $\qquad\square$

**Proposition A.3** (Correctness of GETDEP). *Let* $\mathcal{G}$ *be a causal graph,* $\mathbf{X}, \mathbf{Y}, \mathbf{R}'$ *disjoint sets of variables, and* $\mathbf{T}$ *a set of variables where* $\mathbf{T} \subseteq \mathbf{R}'$. *If there exists a set of variables* $\mathbf{Z}' \subseteq \mathbf{R}' \setminus \mathbf{T}$ *such that* $\mathbf{T} \cup \mathbf{Z}'$ *satisfies the third condition of the FD criterion relative to* $(\mathbf{X}, \mathbf{Y})$, GETDEP *outputs* $\mathbf{Z}'$, *or outputs* $\bot$ *if none exists, in* $O(n^2(n+m))$ *time where* $n$ *and* $m$ *represent the number of nodes and edges in* $\mathcal{G}$.

```
 1: function GETNEIGHBORS(v, 𝒢)
 2:     Output: N all observed neighbors of v in an undirected graph 𝒢.
 3:     N ← observed adjacent nodes of v in 𝒢, mark v and all w ∈ N as visited
 4:     L ← latent adjacent nodes of v in 𝒢, mark all w ∈ L as visited
 5:     Q ← L
 6:     while Q ≠ ∅ do
 7:         u ← Q.POP()
 8:         O ← observed adjacent nodes of u in 𝒢 that have not been visited
 9:         N ← N ∪ O, mark all w ∈ O as visited
10:         L ← latent adjacent nodes of u in 𝒢 that have not been visited
11:         Q.INSERT(L), mark all w ∈ L as visited
12:     end while
13:     return N
14: end function
```

Figure 8: A function that outputs all observed neighbors of a given variable.

*Proof.* GETDEP constructs the graph $\mathcal{G}'$ by starting from the subgraph over $An(\mathbf{T} \cup \mathbf{X} \cup \mathbf{Y})$, and then converting all bidirected edges $A \leftrightarrow B$ into a single latent node $L_{AB}$ and two edges $L_{AB} \to A$ and $L_{AB} \to B$. All outgoing edges of $\mathbf{T}$ are removed from $\mathcal{G}'$ to create $\mathcal{G}''$, which is then moralized to construct an undirected graph $\mathcal{M}$. After, $\mathbf{X}$ is removed from $\mathcal{M}$. The construction of $\mathcal{M}$ is based on the property that $\mathbf{T}$ and $\mathbf{Y}$ are $d$-separated by $\mathbf{X}$ in $\mathcal{G}$ if and only if $\mathbf{X}$ is a $\mathbf{T}$ - $\mathbf{Y}$ node cut (i.e., removing $\mathbf{X}$ disconnects $\mathbf{T}$ from $\mathbf{Y}$) in $\mathcal{G}_0 = \text{MORALIZE}(\mathcal{G}_{An(\mathbf{T} \cup \mathbf{X} \cup \mathbf{Y})})$ [21]. The two tweaks: 1) removing all outgoing edges of $\mathbf{T}$ from $\mathcal{G}'$ before moralization, and 2) removing $\mathbf{X}$ from $\mathcal{M}$ after moralization are added to ensure that all paths from $\mathbf{T}$ to $\mathbf{Y}$ in $\mathcal{M}$ are the BD paths from $\mathbf{T}$ to $\mathbf{Y}$ that cannot be blocked by $\mathbf{X}$ in $\mathcal{G}$.

GETDEP performs BFS from $\mathbf{T}$ to $\mathbf{Y}$ in $\mathcal{M}$. Whenever a node $u$ is visited, GETDEP obtains the non-visited, observed neighbors $\mathbf{NR}$ of $u$ in $\mathcal{M}$ that belong to $\mathbf{R}'$. All observed neighbors of $u$ in $\mathcal{M}$ are obtained by calling the function GETNEIGHBORS($u, \mathcal{M}$) (by Prop. A.2). Then, $\mathcal{M}$ is reconstructed by moralizing the graph $\mathcal{G}'' = \mathcal{G}'_{\overline{\mathbf{T} \cup \mathbf{Z}' \cup \mathbf{NR}}}$ that removes all outgoing edges of $\mathbf{T} \cup \mathbf{Z}' \cup \mathbf{NR}$ from $\mathcal{G}'$, and then removing $\mathbf{X}$ from $\mathcal{M}$ after. All outgoing edges of $\mathbf{NR}$ are removed (in addition to those of $\mathbf{T} \cup \mathbf{Z}'$) to check if removing all outgoing edges of $\mathbf{NR}$ contributes to disconnecting BD paths from $\mathbf{T}$ to $\mathbf{Y}$ that cannot be blocked by $\mathbf{X}$ in $\mathcal{G}$. In other words, $\mathbf{NR}$ may belong to $\mathbf{Z}'$ such that $\mathbf{Z}$ satisfies the third condition of the FD criterion relative to $(\mathbf{X}, \mathbf{Y})$. Hence, $\mathbf{NR}$ is added to $\mathbf{Z}'$.

However, there might exist some BD path $\pi$ from $w \in \mathbf{NR}$ to $y \in \mathbf{Y}$ that cannot be blocked by $\mathbf{X}$ in $\mathcal{G}$. If $\pi$ cannot be disconnected from $w$ to $y$, then $\mathbf{Z}$ will violate the third condition of the FD criterion relative to $(\mathbf{X}, \mathbf{Y})$. We need to check if there exists such $\pi$. GETDEP constructs a set $\mathbf{NR}'$, a set of all variables $w$ in $\mathbf{NR}$ such that there exists an incoming arrow into $w$ in $\mathcal{G}$. Also, there might exist some observed neighbors $\mathbf{N}'$ of $u$ in $\mathcal{M}$ that are still reachable from $u$, even after removing all outgoing edges of $\mathbf{T} \cup \mathbf{Z}' \cup \mathbf{NR}$ (which is reflected by the construction of $\mathcal{M}$). Hence, the union $\mathbf{N}$ of two sets, $\mathbf{N}'$ and $\mathbf{NR}'$, are inserted into $\mathbf{Q}$ to check if any node in $\mathbf{N}$ is reachable to $\mathbf{Y}$.

The BFS continues until either a node $y \in \mathbf{Y}$ is visited, or no more node can be visited. We explain further by each case.

1. A node $y \in \mathbf{Y}$ is visited. There exists no set $\mathbf{Z}'$ such that $\mathbf{Z} = \mathbf{T} \cup \mathbf{Z}'$ satisfies the third condition of the FD criterion relative to $(\mathbf{X}, \mathbf{Y})$. Let $\pi$ be a BD path from $t \in \mathbf{T}$ to $y$ in $\mathcal{M}$ where all nodes in $\pi$ are visited by performing BFS from $t$ to $y$. Since all nodes in $\pi$ are visited, for all variables $w \in \mathbf{R}'$ that intersect $\pi$, all outgoing edges of $w$ must have been removed in $\mathcal{G}''$ and $\mathcal{M}$ was constructed based on $\mathcal{G}''$. However, $y$ was still reached, which implies that removing all outgoing edges of $w$ did not disconnect $\pi$ from $t$ to $y$. Removing all outgoing edges of $\mathbf{R}'$ will not disconnect $\pi$ from $t$ to $y$ either. Thus, there exists no set $\mathbf{Z}'$ such that all BD paths from $\mathbf{Z}$ to $\mathbf{Y}$ are blocked by $\mathbf{X}$ in $\mathcal{G}$. GETDEP returns $\perp$.

2. No more node is left to be visited. All BD paths from $\mathbf{T}$ to $\mathbf{Y}$ that cannot be blocked by $\mathbf{X}$ in $\mathcal{G}$ have been disconnected by removing all outgoing edges of $\mathbf{Z}$ while ensuring that there exists no BD path from $\mathbf{Z}$ to $\mathbf{Y}$ that cannot be blocked by $\mathbf{X}$ in $\mathcal{G}$. All BD paths from $\mathbf{Z}$ to

**Y** are blocked by **X**, and thus **Z** satisfies the third condition of the FD criterion relative to $(\mathbf{X}, \mathbf{Y})$. GETDEP returns the set $\mathbf{Z}'$.

For the time complexity, MORALIZE runs in $O(n^2)$ time. MORALIZE checks over every pair of nodes (of size $O(n^2)$) and adds an undirected edge between each non-adjacent pair if both nodes share a common child. Then, MORALIZE converts all directed edges into undirected edges where the number of edges may be of $O(n^2)$ in the worst case scenario. The BFS takes $O(n^2(n + m))$ time in total since all nodes and edges may be visited at most once (i.e., $O(n + m)$ entities) where visiting a single node takes $O(n^2)$ time where the dominating factor is the runtime of MORALIZE. By Prop. A.2, GETNEIGHBORS runs in $O(n + m)$ time. Hence, GETDEP runs in $O(n^2(n + m))$ time.

$\square$

**Lemma 2** (Correctness of GETCAND3RDFDC). *GETCAND3RDFDC$(\mathcal{G}, \mathbf{X}, \mathbf{Y}, \mathbf{I}, \mathbf{R}')$ in Step 2 of Alg. 1 generates a set of variables $\mathbf{R}''$ where $\mathbf{I} \subseteq \mathbf{R}'' \subseteq \mathbf{R}'$. $\mathbf{R}''$ consists of all and only variables $v$ such that there exists a subset $\mathbf{Z}$ with $\mathbf{I} \subseteq \mathbf{Z} \subseteq \mathbf{R}'$ and $v \in \mathbf{Z}$ that satisfies the third condition of the FD criterion relative to $(\mathbf{X}, \mathbf{Y})$. Further, every set $\mathbf{Z}$ with $\mathbf{I} \subseteq \mathbf{Z} \subseteq \mathbf{R}$ that satisfies both the second and the third conditions of the FD criterion must be a subset of $\mathbf{R}''$.*

*Proof.* The proof consists of two parts.

1. $\mathbf{R}''$ consists of all and only variables $v$ such that there exists a subset $\mathbf{Z}$ with $\mathbf{I} \subseteq \mathbf{Z} \subseteq \mathbf{R}'$ and $v \in \mathbf{Z}$ that satisfies the third condition of the FD criterion relative to $(\mathbf{X}, \mathbf{Y})$.

   GETCAND3RDFDC iterates through all variables $v$ in $\mathbf{R}'$. By Lemma 1, every set $\mathbf{Z}$ with $\mathbf{I} \subseteq \mathbf{Z} \subseteq \mathbf{R}$ that satisfies the second condition of the FD criterion relative to $(\mathbf{X}, \mathbf{Y})$ must be a subset of $\mathbf{R}'$. For each $v$, if GETDEP returns $\perp$, then for every set $\mathbf{Z}$ with $\mathbf{Z} \subseteq \mathbf{R}'$ and $v \in \mathbf{Z}$, $\mathbf{Z}$ violates the third condition of the FD criterion relative to $(\mathbf{X}, \mathbf{Y})$ (by Prop. A.3). Hence, $v$ is removed from $\mathbf{R}''$. All such $v$'s (i.e., $v$ such that GETDEP had returned $\perp$) will be removed from $\mathbf{R}''$. If $v \in \mathbf{I}$, then GETCAND3RDFDC returns $\perp$ as no $\mathbf{Z}$ with $\mathbf{I} \subseteq \mathbf{Z} \subseteq \mathbf{R}'$ and $v \in \mathbf{Z}$ will satisfy the third condition of the FD criterion relative to $(\mathbf{X}, \mathbf{Y})$. At the end of for loop, we have that $\mathbf{R}''$ consists all and only variables $v$ such that there exists a subset $\mathbf{Z}$ with $\mathbf{I} \subseteq \mathbf{Z} \subseteq \mathbf{R}'$ and $v \in \mathbf{Z}$ that satisfies the third condition of the FD criterion relative to $(\mathbf{X}, \mathbf{Y})$.

2. Every set $\mathbf{Z}$ with $\mathbf{I} \subseteq \mathbf{Z} \subseteq \mathbf{R}$ that satisfies both the second and the third conditions of the FD criterion relative to $(\mathbf{X}, \mathbf{Y})$ must be a subset of $\mathbf{R}''$.

   By Lemma 1, every set $\mathbf{Z}$ with $\mathbf{I} \subseteq \mathbf{Z} \subseteq \mathbf{R}$ that satisfies the second condition of the FD criterion relative to $(\mathbf{X}, \mathbf{Y})$ must be a subset of $\mathbf{R}'$. We restrict the scope of $\mathbf{Z}$ into $\mathbf{I} \subseteq \mathbf{Z} \subseteq \mathbf{R}'$ and show that every $\mathbf{Z}$ that satisfies the third condition of the FD criterion relative to $(\mathbf{X}, \mathbf{Y})$ must be a subset of $\mathbf{R}''$.

   When GETCAND3RDFDC iterates through all variables in $\mathbf{R}'$, every $u \in \mathbf{Z}$ must have been checked since $\mathbf{Z} \subseteq \mathbf{R}'$. For each $u \in \mathbf{Z}$, GETDEP must have returned a set of variables since there exists a subset $\mathbf{Z}' = \mathbf{Z} \setminus \{u\} \subseteq \mathbf{R}' \setminus \{u\}$ such that $\mathbf{Z}$ satisfies the third condition of the FD criterion relative to $(\mathbf{X}, \mathbf{Y})$ (by Prop A.3). GETCAND3RDFDC removes all and only variables $v$ from $\mathbf{R}'$ such that there exists no set $\mathbf{Z}'$ with $\mathbf{I} \subseteq \mathbf{Z}' \subseteq \mathbf{R}'$ and $v \in \mathbf{Z}'$ that satisfies the third condition of the FD criterion relative to $(\mathbf{X}, \mathbf{Y})$. If $\mathbf{Z}$ includes any such $v$, then it is a contradiction as $\mathbf{Z}$ will violate the third condition of the FD criterion relative to $(\mathbf{X}, \mathbf{Y})$. Hence, $\mathbf{Z}$ must be a subset of $\mathbf{R}''$.

$\square$

**Proposition A.4** (Complexity of GETCAND3RDFDC). *GETCAND3RDFDC runs in $O(n^3(n + m))$ time where $n$ and $m$ represent the number of nodes and edges in $\mathcal{G}$.*

*Proof.* GETCAND3RDFDC iterates through all variables $v$ in $\mathbf{R}'$ of size $O(n)$. The function GETDEP will be called once per loop. By Prop. A.3, GETDEP runs in $O(n^2(n + m))$ time. In total, the running time of GETCAND3RDFDC is $O(n^3(n + m))$. $\square$

```
1: function GETCAUSALPATHGRAPH($\mathcal{G}, \mathbf{X}, \mathbf{Y}$)
2:     Output: $\mathcal{G}'$ a causal path graph relative to $(\mathcal{G}, \mathbf{X}, \mathbf{Y})$.
3:     $\mathcal{G}'' \leftarrow \mathcal{G}_{\mathbf{X} \cup \mathbf{Y} \cup PCP(\mathbf{X}, \mathbf{Y})}$
4:     $\mathcal{G}' \leftarrow \mathcal{G}''_{\overline{\mathbf{X}} \underline{\mathbf{Y}}}$
5:     Remove all bidirected edges from $\mathcal{G}'$
6:     return $\mathcal{G}'$
7: end function
```

Figure 9: A function that constructs a causal path graph.

**Lemma 3.** $\mathbf{R}''$ *generated by* GETCAND3RDFDC *(in Step 2 of Alg. 1) satisfies the third condition of the FD criterion, that is, all BD paths from* $\mathbf{R}''$ *to* $\mathbf{Y}$ *are blocked by* $\mathbf{X}$.

*Proof.* By Lemma 2, for every variable $v \in \mathbf{R}''$, there exists a subset $\mathbf{Z}' \subseteq \mathbf{R}' \setminus \{v\}$ such that $\mathbf{Z} = \{v\} \cup \mathbf{Z}'$ satisfies the third condition of the FD criterion relative to $(\mathbf{X}, \mathbf{Y})$. In other words, there is no BD path from $\mathbf{Z}$ to $\mathbf{Y}$ that cannot be blocked by $\mathbf{X}$ in $\mathcal{G}$. All BD paths from $v$ to $\mathbf{Y}$ that cannot be blocked by $\mathbf{X}$ are disconnected in $\mathcal{G}_{\underline{\mathbf{Z}}}$ by removing all outgoing edges of $v$ and $\mathbf{Z}'$ in $\mathcal{G}$. Consider the graph $\mathcal{G}_{\underline{\mathbf{R}''}}$ where all outgoing edges of $\mathbf{Z}$ as well as those of $\mathbf{R}'' \setminus \mathbf{Z}$ are removed ($\mathbf{Z} \subseteq \mathbf{R}''$ holds by Lemma 2). Removing more outgoing edges (i.e., in $\mathcal{G}_{\underline{\mathbf{R}''}}$) will not re-connect the BD paths that have already been disconnected in $\mathcal{G}_{\underline{\mathbf{Z}}}$. Hence, all BD paths from $v$ to $\mathbf{Y}$ that cannot be blocked by $\mathbf{X}$ will be disconnected in $\mathcal{G}_{\underline{\mathbf{R}''}}$. Then, for every variable $v \in \mathbf{R}''$, all BD paths from $v$ to $\mathbf{Y}$ that cannot be blocked by $\mathbf{X}$ will be disconnected in $\mathcal{G}_{\underline{\mathbf{R}''}}$. All BD paths from $\mathbf{R}''$ to $\mathbf{Y}$ that cannot be blocked by $\mathbf{X}$ are disconnected in $\mathcal{G}_{\underline{\mathbf{R}''}}$. All BD paths from $\mathbf{R}''$ to $\mathbf{Y}$ are blocked by $\mathbf{X}$ and thus $\mathbf{R}''$ satisfies the third condition of the FD criterion relative to $(\mathbf{X}, \mathbf{Y})$.

$\square$

**Proposition A.5.** *Let* $\mathcal{G}$ *be a causal graph and* $\mathbf{X}, \mathbf{Y}$ *disjoint sets of variables.* GETCAUSALPATH-GRAPH *constructs a causal path graph* $\mathcal{G}'$ *relative to* $(\mathcal{G}, \mathbf{X}, \mathbf{Y})$ *in* $O(n + m)$ *time where* $n$ *and* $m$ *represent the number of nodes and edges in* $\mathcal{G}$.

*Proof.* The construction of a causal path graph is immediate from Def. 2. Constructing a subgraph $\mathcal{G}_{\mathbf{X} \cup \mathbf{Y} \cup PCP(\mathbf{X}, \mathbf{Y})}$, performing graph transformation $\mathcal{G}''_{\overline{\mathbf{X}} \underline{\mathbf{Y}}}$, and removing all bidirected edges take $O(n + m)$ time. $\square$

**Definition 3.** (Proper Causal Path [35]) Let $\mathbf{X}, \mathbf{Y}$ be set of nodes. A causal path from a node in $\mathbf{X}$ to a node in $\mathbf{Y}$ is called proper if it does not intersect $\mathbf{X}$ except at the end point.

**Lemma 4.** *Let* $\mathcal{G}$ *be a causal graph and* $\mathbf{X}, \mathbf{Y}, \mathbf{Z}$ *disjoint sets of variables. Let* $\mathcal{G}'$ *be the causal path graph relative to* $(\mathcal{G}, \mathbf{X}, \mathbf{Y})$. *Then,* $\mathbf{Z}$ *satisfies the first condition of the FD criterion relative to* $(\mathbf{X}, \mathbf{Y})$ *if and only if* $\mathbf{Z}$ *is a separator of* $\mathbf{X}$ *and* $\mathbf{Y}$ *in* $\mathcal{G}'$.

*Proof.* We prove the statement in both directions.

- *If case:* We show that $\mathbf{Z}$ satisfies the first condition of the FD criterion relative to $(\mathbf{X}, \mathbf{Y})$. By the construction of $\mathcal{G}'$, all paths from $\mathbf{X}$ to $\mathbf{Y}$ comprise of all and only proper causal paths from $\mathbf{X}$ to $\mathbf{Y}$. It is only necessary to check for all proper causal paths from $\mathbf{X}$ to $\mathbf{Y}$ since every non-proper causal path from $\mathbf{X}$ to $\mathbf{Y}$ must include a proper causal path from $\mathbf{X}$ to $\mathbf{Y}$ as a subpath. To witness, consider any non-proper causal path $\pi = x_1 \to , \cdots , \to x_k \to , \cdots , \to y$ from a node $x_1 \in \mathbf{X}$ to a node $y \in \mathbf{Y}$. Since $\pi$ is not proper, there must exist a node $x_k \in \mathbf{X}$ that intersects $\pi$ at non-endpoint and there exists a subpath $\pi' = x_k \to , \cdots , \to y$ such that $\pi'$ is proper. Since $\mathbf{Z}$ is a separator of $\mathbf{X}$ and $\mathbf{Y}$ in $\mathcal{G}'$, $\mathbf{Z}$ intercepts all causal paths from $\mathbf{X}$ to $\mathbf{Y}$ in $\mathcal{G}$.

- *Only if case:* We show that $\mathbf{Z}$ is a separator of $\mathbf{X}$ and $\mathbf{Y}$ in $\mathcal{G}'$. By assumption, $\mathbf{Z}$ intercepts all causal paths from $\mathbf{X}$ to $\mathbf{Y}$ in $\mathcal{G}$. By the construction of $\mathcal{G}'$, all and only paths from $\mathbf{X}$ to $\mathbf{Y}$ must be causal. Thus, $\mathbf{Z}$ must be a separator of $\mathbf{X}$ and $\mathbf{Y}$ in $\mathcal{G}'$.

$\square$

**Lemma 5.** *There exists a set $\mathbf{Z}_0$ satisfying the FD criterion relative to $(\mathbf{X}, \mathbf{Y})$ with $\mathbf{I} \subseteq \mathbf{Z}_0 \subseteq \mathbf{R}$ if and only if $\mathbf{R}''$ generated by* GETCAND3RDFDC *(in Step 2 of Alg. 1) satisfies the FD criterion relative to $(\mathbf{X}, \mathbf{Y})$.*

We prove the statement in both directions.

- *If case:* It is automatic with $\mathbf{Z}_0 = \mathbf{R}''$.

- *Only if case:* We prove the contrapositive of the statement: if $\mathbf{R}''$ is not a FD adjustment set relative to $(\mathbf{X}, \mathbf{Y})$, then there does not exist any FD adjustment set $\mathbf{Z}_0$ relative to $(\mathbf{X}, \mathbf{Y})$ with $\mathbf{I} \subseteq \mathbf{Z}_0 \subseteq \mathbf{R}$. On the following three items, we show that there does not exist any FD adjustment set $\mathbf{Z}_0$ relative to $(\mathbf{X}, \mathbf{Y})$ with three disjoint intervals, $\mathbf{I} \subseteq \mathbf{Z}_0 \subseteq \mathbf{R}''$, $\mathbf{R}'' \subset \mathbf{Z}_0 \subseteq \mathbf{R}'$, and $\mathbf{R}' \subset \mathbf{Z}_0 \subseteq \mathbf{R}$, respectively.

  1. Since $\mathbf{R}''$ is not a FD adjustment set relative to $(\mathbf{X}, \mathbf{Y})$, $\mathbf{R}''$ must be violating the first condition of the FD criterion relative to $(\mathbf{X}, \mathbf{Y})$. That is because, by the construction of $\mathbf{R}''$, $\mathbf{R}''$ must satisfy the second condition of the FD criterion relative to $(\mathbf{X}, \mathbf{Y})$ (by Lemma 1) and the third condition of the FD criterion relative to $(\mathbf{X}, \mathbf{Y})$ (by Lemma 3). Then, $\mathbf{R}''$ does not intercept all causal paths from $\mathbf{X}$ to $\mathbf{Y}$. No subset $\mathbf{Z}_0$ with $\mathbf{I} \subseteq \mathbf{Z}_0 \subseteq \mathbf{R}''$ will intercept all causal paths from $\mathbf{X}$ to $\mathbf{Y}$. $\mathbf{Z}_0$ violates the first condition of the FD criterion relative to $(\mathbf{X}, \mathbf{Y})$), and thus $\mathbf{Z}_0$ is not a FD adjustment set relative to $(\mathbf{X}, \mathbf{Y})$.

  2. Consider a collection of sets $\mathbf{Z}_0$ with $\mathbf{R}'' \subset \mathbf{Z}_0 \subseteq \mathbf{R}'$. By the construction of $\mathbf{R}''$ generated by GETCAND3RDFDC (with $\mathbf{R}'' \subseteq \mathbf{R}'$), for all $v \in \mathbf{R}' \setminus \mathbf{R}''$, there does not exist any set $\mathbf{Z}$ with $\mathbf{I} \subseteq \mathbf{Z} \subseteq \mathbf{R}'$ and $v \in \mathbf{Z}$ that satisfies the third condition of the FD criterion relative to $(\mathbf{X}, \mathbf{Y})$ (by Lemma 2). $\mathbf{Z}_0$ must include some $v$, and thus $\mathbf{Z}_0$ violates the third condition of the FD criterion relative to $(\mathbf{X}, \mathbf{Y})$. $\mathbf{Z}_0$ is not a FD adjustment set relative to $(\mathbf{X}, \mathbf{Y})$.

  3. Consider a collection of sets $\mathbf{Z}_0$ with $\mathbf{R}' \subset \mathbf{Z}_0 \subseteq \mathbf{R}$. By the construction of $\mathbf{R}'$ generated by GETCAND2NDFDC (with $\mathbf{R}' \subseteq \mathbf{R}$), for all $v \in \mathbf{R} \setminus \mathbf{R}'$, there exists an open BD path from $\mathbf{X}$ to $v$ (By Lemma 1). $\mathbf{Z}_0$ must be including some $v$, and by definition, there exists an open BD path from $\mathbf{X}$ to $\mathbf{Z}_0$. $\mathbf{Z}_0$ violates the second condition of the FD criterion relative to $(\mathbf{X}, \mathbf{Y})$ and $\mathbf{Z}_0$ is not a FD adjustment set relative to $(\mathbf{X}, \mathbf{Y})$.

  Combining together the three items, we have that for all $\mathbf{Z}_0$ with $\mathbf{I} \subseteq \mathbf{Z}_0 \subseteq \mathbf{R}$, $\mathbf{Z}_0$ is not a FD adjustment set relative to $(\mathbf{X}, \mathbf{Y})$.

**Theorem 1** (Correctness of FINDFDSET). *Let $\mathcal{G}$ be a causal graph, $\mathbf{X}, \mathbf{Y}$ disjoint sets of variables, and $\mathbf{I}, \mathbf{R}$ sets of variables such that $\mathbf{I} \subseteq \mathbf{R}$. Then,* FINDFDSET$(\mathcal{G}, \mathbf{X}, \mathbf{Y}, \mathbf{I}, \mathbf{R})$ *outputs a set $\mathbf{Z}$ with $\mathbf{I} \subseteq \mathbf{Z} \subseteq \mathbf{R}$ that satisfies the FD criterion relative to $(\mathbf{X}, \mathbf{Y})$, or outputs $\perp$ if none exists, in $O(n^3(n+m))$ time, where $n$ and $m$ represent the number of nodes and edges in $\mathcal{G}$.*

*Proof.* By Lemma 2, every set $\mathbf{Z}$ with $\mathbf{I} \subseteq \mathbf{Z} \subseteq \mathbf{R}$ that satisfies both the second and the third conditions of the FD criterion relative to $(\mathbf{X}, \mathbf{Y})$ must be a subset of $\mathbf{R}''$. By Lemma 1, $\mathbf{R}''$ satisfies the second condition of the FD criterion relative to $(\mathbf{X}, \mathbf{Y})$. By Lemma 3, $\mathbf{R}''$ satisfies the third condition of the FD criterion relative to $(\mathbf{X}, \mathbf{Y})$. Let $\mathbf{Z} = \mathbf{R}''$. Then, By Lemma 4, $\mathbf{Z}$ is a FD adjustment set relative to $(\mathbf{X}, \mathbf{Y})$ if and only if $\mathbf{Z}$ is a separator of $\mathbf{X}$ and $\mathbf{Y}$ in $\mathcal{G}'$, a causal path graph relative to $(\mathcal{G}, \mathbf{X}, \mathbf{Y})$. FINDFDSET outputs $\mathbf{Z}$ if and only if $\mathbf{Z}$ is a separator of $\mathbf{X}$ and $\mathbf{Y}$ in $\mathcal{G}'$ (by calling TESTSEP$(\mathcal{G}', \mathbf{X}, \mathbf{Y}, \mathbf{Z})$ at line 11 and verifying TESTSEP is returning True). Hence, the outputted set $\mathbf{Z}$ is a FD adjustment set relative to $(\mathbf{X}, \mathbf{Y})$ where $\mathbf{I} \subseteq \mathbf{Z} \subseteq \mathbf{R}$. If TESTSEP returns False, then $\mathbf{Z}$ is not a FD adjustment set relative to $(\mathbf{X}, \mathbf{Y})$ and FINDFDSET outputs $\perp$. By Lemma 5, there does not exist any FD adjustment set $\mathbf{Z}_0$ relative to $(\mathbf{X}, \mathbf{Y})$ with $\mathbf{I} \subseteq \mathbf{Z}_0 \subseteq \mathbf{R}$.

For the running time, constructing $\mathbf{R}'$ takes $O(n(n+m))$ time (by Prop. A.1), and generating $\mathbf{R}''$ takes $O(n^3(n+m))$ time (by Prop. A.4). By Prop. A.5, creating a causal path graph $\mathcal{G}'$ relative to $(\mathcal{G}, \mathbf{X}, \mathbf{Y})$ (by calling GETCAUSALPATHGRAPH) takes $O(n+m)$ time. TESTSEP takes $O(n+m)$ time. The dominant factor is $O(n^3(n+m))$. $\qquad\square$

**Theorem 2** (Correctness of LISTFDSETS). *Let $\mathcal{G}$ be a causal graph, $\mathbf{X}, \mathbf{Y}$ disjoint sets of variables, and $\mathbf{I}, \mathbf{R}$ sets of variables.* LISTFDSETS$(\mathcal{G}, \mathbf{X}, \mathbf{Y}, \mathbf{I}, \mathbf{R})$ *enumerates all and only sets $\mathbf{Z}$ with $\mathbf{I} \subseteq \mathbf{Z} \subseteq \mathbf{R}$ that satisfy the FD criterion relative to $(\mathbf{X}, \mathbf{Y})$ in $O(n^4(n+m))$ delay where $n$ and $m$ represent the number of nodes and edges in $\mathcal{G}$.*

*Proof.* Consider the recursion tree for LISTFDSETS. By induction on tree nodes, we show that when a tree node $\mathcal{N}(\mathbf{I}', \mathbf{R}')$ is visited, LISTFDSETS will output all and only FD adjustment sets $\mathbf{Z}$ relative to $(\mathbf{X}, \mathbf{Y})$ where $\mathbf{I}' \subseteq \mathbf{Z} \subseteq \mathbf{R}'$.

- *Base case*: Consider any leaf tree node $\mathcal{L}(\mathbf{I}', \mathbf{R}')$. The recursion stops when $\mathbf{I} = \mathbf{R}$, so $\mathbf{I}' = \mathbf{R}'$ must hold. $\mathcal{L}$ contains a node $\mathbf{Z}$ with $\mathbf{Z} = \mathbf{I}' = \mathbf{R}'$ if $\mathbf{Z}$ is a valid FD adjustment set relative to $(\mathbf{X}, \mathbf{Y})$, or empty otherwise. Indeed, LISTFDSETS will output a FD adjustment set $\mathbf{Z}$ if and only if FINDFDSET in line 3 does not output $\perp$ (by Thm. 1).

- *Inductive case*: Let $\mathcal{N}(\mathbf{I}', \mathbf{R}')$ be any non-leaf tree node. Assume the claim holds for two children of $\mathcal{N}$. We show that $\mathcal{N}$ contains all FD adjustment sets $\mathbf{Z}$ with $\mathbf{I}' \subseteq \mathbf{Z} \subseteq \mathbf{R}'$, which can also be expressed as the union of two collections of sets: 1) the collection of FD adjustment sets $\mathbf{Z}_1$ with $\mathbf{I}' \cup \{v\} \subseteq \mathbf{Z}_1 \subseteq \mathbf{R}'$, and 2) the collection of FD adjustment sets $\mathbf{Z}_2$ with $\mathbf{I}' \subseteq \mathbf{Z}_2 \subseteq \mathbf{R}' \setminus \{v\}$. The two collections are disjoint as every set in the first collection contains $v$, and none in the second collection does. By assumption, each child contains the collection of respective FD adjustment sets. If FINDFDSET in line 3 outputs $\perp$, then there does not exist a FD adjustment set $\mathbf{Z}$ with $\mathbf{I}' \subseteq \mathbf{Z} \subseteq \mathbf{R}'$. Otherwise, each child outputs a respective collection of FD adjustment sets.

For the runtime, consider the recursion tree for LISTFDSETS. Every time a tree node $\mathcal{N}(\mathbf{I}', \mathbf{R}')$ is visited, the function FINDFDSET is called, which takes $O(n^3(n + m))$ time (by Thm. 1). If FINDFDSET outputs $\perp$, then LISTFDSETS does not search further from $\mathcal{N}$ because there exists no FD adjustment set $\mathbf{Z}$ with $\mathbf{I}' \subseteq \mathbf{Z} \subseteq \mathbf{R}'$. Otherwise, recursion continues until a leaf tree node is visited. In each level of the tree, a single node $v$ is removed from the set $\mathbf{R} \setminus \mathbf{I}$. The depth of the tree is at most $n$, and the time required to output a set $\mathbf{Z}$ is $O(n^4(n + m))$. In the worst case scenario, $n$ branches will be aborted (i.e., FINDFDSET outputs $\perp$ on every level of the tree) before reaching the first leaf. It takes $O(n^4(n + m))$ time to produce either the first output or halt. Thus, LISTFDSETS runs with $O(n^4(n + m))$ delay. $\square$