# OpenReview forum: "Finding and Listing Front-door Adjustment Sets"
_NeurIPS.cc/2022/Conference — NeurIPS 2022 Accept_

### Official Review · Reviewer_ebQn · 2022-06-25

**Rating:** 7
**Confidence:** 4
**Soundness:** 2 fair
**Presentation:** 3 good
**Contribution:** 3 good

**Summary:**

The paper develops an algorithm, i.e. FINDFDSE() to find an admissible adjustment set that satisfies the front-door criterion in a given causal DAG. The solution is to solve a variant of the problem that imposes constraints I for given sets I and R.  Based on the algorithm FINDFDSE(), a  sound and complete algorithm is developed to enumerate all front-door adjustment sets with polynomial time.  Moreover, the detailed theoretical analysis supports the soundness of the developed LISTFDSETS() algorithm.


**Questions:**

Q1. is the first weakness.

Q2. In Fig. 1b, If the edge $A\rightarrow D$ is replaced by $A\leftarrow D$, is there still an admission front-door adjustment set $\mathbf{Z]$? If the answer is no, are there some graphical criteria to handle such a case?

**Limitations:**

1. The causal DAG and the sets $\mathbf{I}$ and $\mathbf{R}$  need to be provided. In many real-world applications, this information is not available. Hence, the application of the developed algorithms in this work has limited application at present.

2. For multiple treatments and outcomes $(\mathbf{X, Y})$, the developed algorithms may only be able to solve very few cases, and most of them return $\perp$.

**Strengths And Weaknesses:**

*Strengths:
1. This work is well-motivated and organised. A large number of examples make the paper easy to follow and understand.
2. The technical quality of the work is high. The steps of both algorithms are also very clear.

*Weaknesses:
1. Assumptions are not clear. To my understanding, the given causal DAG is a very specific class of DAGs. For example, $\mathbf{X}=\\{X_1, X_2\\}$, $\mathbf{Y}=\\{Y\\}$, $\mathbf{R}=\\{A,B,C\\}$; the graph is formed: $Y\leftrightarrow X_1 \rightarrow A\rightarrow B\rightarrow Y$; $Y\leftrightarrow X_2 \leftarrow A$; $B\rightarrow C \leftarrow X_2$; $C\rightarrow Y$. $\forall v\in \mathbf{R}$, we have TESTSEP($\mathcal{G}_{\underline{\mathbf{X}}}$, $\mathbf{X}$, $v$, $\emptyset$) = FALSE. Thus, FINDFDSET() returns $\perp$. However, for a single $X\in \mathbf{X}$,   FINDFDSET() returns $A$ for $(X_1, Y)$ and $C$ for $(X_2, Y)$, respectively. Therefore, the generalization on $\mathbf{X}$ may not be perfect.

2. The ``completeness'' is not specified. Furthermore, there is no evidence for completeness.

Some minors:
1. Definition 1, "There is no back-door path from $\mathbf{X}$ to $\mathbf{Z}$"  should be "There is no unblocked back-door path from $\mathbf{X}$ to $\mathbf{Z}$".

2. $\mathbf{PA}\_{V}$ and $\mathbf{pa}\_{V}$ are not defined.

---

> ### Author Response · Authors · 2022-08-01
> **Response to Reviewer ebQn - Part 2**
>
> > For multiple treatments and outcomes (X,Y), the developed algorithms may only be able to solve very few cases, and most of them return \perp.
>
> In cases when FindFDSet (Algorithm 1) returns $\perp$, either P(y | do(x)) is not identifiable at all, or P(y | do(x)) is identifiable but not by Pearl’s FD criterion, which only covers a fraction of the identifiable cases. Your observation could be due to that, for multiple treatments and outcomes, P(y | do(x)) is often not identifiable, or P(y | do(x)) is identifiable but often not by Pearl’s FD criterion.
>
> We emphasize that Algorithms 1 and 2 are complete for multiple treatments and outcomes in the sense that FindFDSet will output a FD adjustment set if one exists and ListFDSets will list all of the sets Z that satisfy the FD criterion. They will return $\perp$ only when no FD sets exist.

---

> > ### Comment · Reviewer_ebQn · 2022-08-09
> > **Response to clarify**
> >
> > Thanks for your clarification. I have understood the meaning of ``completeness'' used in this work.
> >
> > It would be better to add a couple of sentences to make this point clearly in the main text.
> >
> > As a result, I am happy to increase my score.

---

> ### Author Response · Authors · 2022-08-01
> **Response to Reviewer ebQn - Part 1**
>
> We thank the reviewer for thoughtful feedback.
>
> > Q1. Assumptions are not clear. To my understanding, the given causal DAG is a very specific class of DAGs. For example, X={X1,X2}, Y={Y}, R={A,B,C}; the graph is formed: Y↔X1→A→B→Y; Y↔X2←A; B→C←X2; C→Y. ∀v∈R, we have TESTSEP($G_{\underline{\text{X}}}$, X, v, ∅) = FALSE. Thus, FINDFDSET() returns ⊥. However, for a single X∈X, FINDFDSET() returns A for (X1,Y) and C for (X2,Y), respectively.Therefore, the generalization on X may not be perfect.
>
> In this graph, for X = {X1, X2}, FindFDSet() returns $\perp$; For X = {X1}, FindFDSet() actually returns {A,B} although {A} is also a valid FD adjustment set; For X = {X2}, FindFDSet() actually returns $\perp$ because there is a BD path $X2 \leftarrow A \rightarrow B \rightarrow C$ such that no sets satisfy the second condition of the FD criterion.
>
> We do not make any assumption about specific classes of DAGs. In general, there is no known connection between the FD adjustment sets for identifying P(y | do(x1, x2)) and the FD adjustment sets for identifying P(y | do(x1)) or P(y | do(x2)). For concreteness, consider the following example: $G = X1 \rightarrow A \rightarrow Y; X2 \rightarrow B \rightarrow Y; A \rightarrow B; X2 \leftrightarrow X1 \leftrightarrow Y$ with I = $\emptyset$ and R = {A,B}. For X = {X1}, {A} is a valid FD adjustment set. For X = {X2}, there exists none. However, for X = {X1,X2}, {A,B} is a valid FD adjustment set.
>
> > The ``completeness'' is not specified. Furthermore, there is no evidence for completeness.
>
> The ‘completeness’ of ListFDSets (Algorithm 2) means that it will list *all* sets $Z$ with $I \subseteq Z \subseteq R$ that satisfy the FD criterion. This is formally established in Theorem 2. On the other hand, as mentioned in Section 5, page 9, line 386-388, Pearl’s FD criterion is not complete, that is, there might exist a causal effect that can be computed by the FD adjustment formula but the set $Z$ does not satisfy the FD criterion.
>
> > Q2. In Fig. 1b, If the edge A -> D is replaced by A <- D, is there still an admission front-door adjustment set $\mathbf{Z}$? If the answer is no, are there some graphical criteria to handle such a case?
>
> In this case, there exists no FD adjustment set relative to (X,Y) due to the BD path $X \leftrightarrow D \rightarrow A$. However, the causal effect P(y | do(x)) is still identifiable (albeit not via either BD or FD criteria), for example, by a graphical criterion in [28, Tian and Pearl 2002] that says, for a singleton X, ‘P(v | do(x)) is identifiable if and only if there is no bidirected path connecting X to any of its children’. In general, algorithms [7, 22] have been developed to determine the identifiability of interventional distributions in the form of P(y | do(x)).
>
> > The causal DAG and the sets $I$ and $R$ need to be provided. In many real-world applications, this information is not available. Hence, the application of the developed algorithms in this work has limited application at present.
>
> We respectfully note that there is a misunderstanding of how the algorithm works, and ‘the sets $I$ and $R$ need to be provided’ is more a convenient feature than a limitation. Giving users the option of providing $I$ and $R$ allows one to put constraints on candidate adjustment sets based on practical considerations, e.g., avoid adjusting for undesirable variables by excluding them from $R$. Still, by default, one can always use $I = \emptyset$ and $R = V \setminus (X \cup Y)$, where $V$ represents all observed variables in the model, which then puts no restrictions on candidate adjustment sets. Furthermore, two sets $I$ and $R$ allow FindFDSet to function as a building block for the algorithm (ListFDSets) that enumerates all valid FD adjustment sets $Z$ with $I \subseteq Z \subseteq R$. ListFDSets utilizes the result obtained by FindFDSet during the recursive call. The motivation for introducing $I$ and $R$ is explained in the first paragraph of Section 3, lines 135-156.

---

### Official Review · Reviewer_tJUN · 2022-07-07

**Rating:** 7
**Confidence:** 4
**Soundness:** 4 excellent
**Presentation:** 4 excellent
**Contribution:** 3 good

**Summary:**

Given the underlying causal diagram, this paper provides two algorithms:
1. To find an admissible front-door adjustment set in polynomial time (if one exists).
2. To enumerate all admissible front-door adjustment sets with polynomial delay.

These algorithms allow constraining the search to include and/or exclude specific subsets of variables which could help selecting estimands with desired properties.

**Questions:**

Suggestions:
1. As mentioned by authors in section 1 (line 77) and in section 5 (lines 393-395), there has been a lot of work on finding/listing back-door admissible sets when the underlying DAG/MAG/PAG/CPDAG is known. More importantly, there has also been some work on data-driven approaches for finding/listing back-door admissible sets using only an anchor variable when the underlying causal diagram is not known. For example: (1) Entner et al 2013 -- Data-driven covariate selection for nonparametric estimation of causal effects [https://proceedings.mlr.press/v31/entner13a.html], (2) Cheng et al 2020 -- Towards unique and unbiased causal effect estimation from data with hidden variable [https://arxiv.org/pdf/2002.10091.pdf], (3) Shah et al 2021 -- Finding Valid Adjustments under Non-ignorability with Minimal DAG Knowledge [https://arxiv.org/pdf/2106.11560.pdf]. In order to provide a comprehensive background regarding what is know for finding/listing back-door admissible sets, the authors should also talk about these works.
2. As mentioned in the weaknesses section above, this paper assumes the knowledge of the underlying causal diagram in order to find admissible front-door adjustment sets. In contrast, the recent work of Bhattacharya and Nabi-- On Testability of the Front-Door Model via Verma Constraints [https://arxiv.org/pdf/2203.00161v1.pdf] -- explores finding admissible front-door adjustment sets in a data-driven manner using only an anchor variable when the underlying causal structure is not known similar to the works mentioned above. The authors should differentiate their work from this work.

Typo:
1. Did the authors intend to say $3^n$ instead of $2^n$ on line 339?


**Limitations:**

In my opinion, the most important limitation of this work is the requirement of the underlying causal diagram which is acknowledged by the authors in Section 5. This limits the application of this work in most real-world scenarios where the causal diagram is not known.

Therefore, in addition to finding/listing front-door admissible sets from incomplete or partially specified causal diagram as mentioned by the authors (in lines 395-396), it would be useful (for future research) to develop data-driven methods when the underlying causal structure is not known (except an anchor variable) by taking inspiration from the works of Entner et al 2013, Shah et al 2022, Cheng et al 2022, and Bhattacharya and Nabi 2022.

**Strengths And Weaknesses:**

Strength:
1. In my limited knowledge, this paper is one of the first papers to provide algorithms to find/list all admissible front-door adjustment sets (given the underlying causal diagram).
2. Finding valid front-door sets is an important topic and of relevance to the NeurIPS community. Through this work, the authors take a first step to close the gap between what is known for back-door adjustment sets and what is known for front-door adjustment sets.
3. By constraining the search space to include/exclude specific subsets of variables, the authors provide flexibility in selecting estimands with different cost, availability, privacy, and statistical power.

Originality:
1. The algorithm/approach proposed in the paper relies on a mix of novel concepts (e.g., GETDEP, GETNEIGHBORS) as well as ideas known in the literature (e.g., TESTSEP, LISTSEP). However, the key novelty lies in unifying these concepts to propose FINDFDSET and LISTFDSETS.

Weakness:
1. This paper assumes the knowledge of the underlying causal diagram in order to find admissible front-door adjustment sets. In most real-world applications, the causal diagrams are unknown and this requirement is quite restrictive.

Clarity:
1. In general, the paper is quite well written. Further, the algorithms and the functions are generally well explained in the text. While the authors do a decent job, the function GETDEP is a bit complicated to understand. I strongly recommend the authors to add running examples covering various scenarios that may arise in this function to make it easier for the readers.

---

> ### Author Response · Authors · 2022-08-01
> **Response to Reviewer tJUN**
>
> We thank the reviewer for the time reviewing our paper and valuable feedback provided.
>
> > As mentioned by authors in section 1 (line 77) and in section 5 (lines 393-395), there has been a lot of work on finding/listing back-door admissible sets when the underlying DAG/MAG/PAG/CPDAG is known. More importantly, there has also been some work on data-driven approaches for finding/listing back-door admissible sets using only an anchor variable when the underlying causal diagram is not known. For example: (1) Entner et al 2013 -- Data-driven covariate selection for nonparametric estimation of causal effects [https://proceedings.mlr.press/v31/entner13a.html], (2) Cheng et al 2020 -- Towards unique and unbiased causal effect estimation from data with hidden variable [https://arxiv.org/pdf/2002.10091.pdf], (3) Shah et al 2021 -- Finding Valid Adjustments under Non-ignorability with Minimal DAG Knowledge [https://arxiv.org/pdf/2106.11560.pdf]. In order to provide a comprehensive background regarding what is know for finding/listing back-door admissible sets, the authors should also talk about these works.
>
> We will add a discussion of this line of work and the corresponding citations in Section 5, thank you for the suggestion.
>
> > As mentioned in the weaknesses section above, this paper assumes the knowledge of the underlying causal diagram in order to find admissible front-door adjustment sets. In contrast, the recent work of Bhattacharya and Nabi-- On Testability of the Front-Door Model via Verma Constraints [https://arxiv.org/pdf/2203.00161v1.pdf] -- explores finding admissible front-door adjustment sets in a data-driven manner using only an anchor variable when the underlying causal structure is not known similar to the works mentioned above. The authors should differentiate their work from this work.
>
> Thank you for pointing out this work. We will discuss this work in Section 5.
>
> > Typo: Did the authors intend to say 3^n instead of 2^n on line 339?
>
> You are right, thanks for the catch.

---

> > ### Comment · Reviewer_tJUN · 2022-08-08
> > **Re paper clarity**
> >
> > I would encourage the authors to add examples covering various scenarios that may arise in the GETDEP function in the revision.

---

> > > ### Author Response · Authors · 2022-08-09
> > > **Reply to Re paper clarity**
> > >
> > > Thank you for the note. We certainly agree and appreciate it, and will implement it accordingly.

---

### Official Review · Reviewer_yYRf · 2022-07-10

**Rating:** 6
**Confidence:** 4
**Soundness:** 4 excellent
**Presentation:** 4 excellent
**Contribution:** 4 excellent

**Summary:**

The authors propose a feasible algorithm for listing adjustment sets (one at a time in reasonable time) satisfying the front-door criterion.

**Questions:**

I did have some questions after having read the paper, which I didn't know how to answer.

* For medical or ethical reasons, it may be necessary to avoid including certain variables in an adjustment set. Is this straightforward to do with this algorithm (i.e., simply not including the prohibited variables in any set in the course of the algorithm), or could it possibly be the case the simply excluding certain variable will prevent allowable front-door adjustment sets from being discovered?

* For ethical reasons (fairness, for example), it may be necessary to disallow certain paths of influence in a model (e.g., race affecting the outcome variable). Can this be done inside the algorithms, or some one have to simply list example front-door adjustment sets and simply ignore the offending examples?

**Limitations:**

The last sentence addresses societal impacts: "We hope that the methods and algorithms proposed in this work will help scientists to use the FD strategy for causal effects estimation in the practical applications and are useful for scientists in study design to select covariates based on desired properties, including cost, feasibility, and statistical power." This can of course be made more than a hope; perhaps answering questions such as the above (or others) might help make that a reality.

**Strengths And Weaknesses:**

This is very strong paper, very clearly written and easy to follow. The theory and algorithms are clear and make sense. It should be easy to implement the algorithm described. The idea is novel, since no such algorithm currently exists in the literature, and it is significant, since it could easily be incorporated into existing software and used. The only criticism is that it does not include an experimental section in which the algorithm is test for accuracy and speed on particular models.

Minor point.

Line 252, "no more node" should be "no more nodes".

---

> ### Author Response · Authors · 2022-08-01
> **Response to Reviewer yYRf**
>
> We thank the reviewer for valuable feedback.
>
> > For medical or ethical reasons, it may be necessary to avoid including certain variables in an adjustment set. Is this straightforward to do with this algorithm (i.e., simply not including the prohibited variables in any set in the course of the algorithm), or could it possibly be the case the simply excluding certain variable will prevent allowable front-door adjustment sets from being discovered?
>
> Yes, this is natural and straightforward to do with our strategy, and can be accomplished by simply not including the prohibited variables in $R$ in Algorithms 1 and 2.
>
> > For ethical reasons (fairness, for example), it may be necessary to disallow certain paths of influence in a model (e.g., race affecting the outcome variable). Can this be done inside the algorithms, or some one have to simply list example front-door adjustment sets and simply ignore the offending examples?
>
> If we understand the question and the idea of 'disallow certain paths of influence in a model' correctly, we don’t think it’s possible to easily perform such a task. If we want to discount the effect of some paths, a more refined notion of change involving counterfactuals and path-specific effects is required [Pearl 2001], but our goal here is to identify interventional distribution in the form P(y | do(x)).
>
> [Pearl 2001] Judea Pearl. Direct and Indirect Effects. In Proceedings of the Seventeenth Conference on Uncertainty in Artificial Intelligence, San Francisco, CA: Morgan Kaufmann, 411-420, 2001.

---

> > ### Comment · Reviewer_yYRf · 2022-08-09
> > **Thanks.**
> >
> > I see. That makes sense.

---

### Official Review · Reviewer_879A · 2022-07-12

**Rating:** 8
**Confidence:** 4
**Soundness:** 3 good
**Presentation:** 3 good
**Contribution:** 3 good

**Summary:**

This paper shows the more detailed steps and pseudo-codes for finding frond-door adjustment sets in causal graphical models. Starting from the definition of front door criteria, this paper goes through the implementation of each criterion and offers a complexity analysis.

**Questions:**

I think the complex statements are better to be shown in the main text.


This algorithm works by removing nodes from the full set of candidates, which could be all nodes except for the X and Y.
Does this algorithm take into account "evidence variables"?
If we find a full list of FD sets from a causal graphical model, and if we introduce evidence variables, should we restart from the beginning?


Have you implemented these pseudo-codes?
I think it's better to double-check it by implementation on top of proofs and hand-written examples.
It is in a polynomial-time algorithm in terms of big-O analysis, but the overall running time may be longer than one might imagine when there are many nodes and edges in the graph. This algorithm can be served as a baseline for possible future works that improve running times and may be useful for implementing do-calculus. They are very close to script languages such as python already, and if possible, showing the results through running the program or providing it as open-source could be even more helpful to scientists who may use this FD strategy for experiment design.



**Limitations:**

I think this work is not relevant to this section.

**Strengths And Weaknesses:**

The strength of this paper is showing the detailed steps for implementing frond door criteria in causal graphical models, and it finally extends by listing all frond door sets satisfying Pearl's front door criteria.
The weakness is that no demonstration or evaluation is available for given algorithms.

---

> ### Author Response · Authors · 2022-08-01
> **Response to Reviewer 879A**
>
> We thank the reviewer for the time and the valuable feedback provided.
>
> > This algorithm works by removing nodes from the full set of candidates, which could be all nodes except for the X and Y. Does this algorithm take into account "evidence variables"? If we find a full list of FD sets from a causal graphical model, and if we introduce evidence variables, should we restart from the beginning?
>
> The current algorithm is for marginal effects, P(y| do(x)), and does not take into account evidence variables. We mentioned finding conditional FD sets as a future work in Section 5, page 9, line 396-399.
>
> > Have you implemented these pseudo-codes? I think it's better to double-check it by implementation on top of proofs and hand-written examples. It is in a polynomial-time algorithm in terms of big-O analysis, but the overall running time may be longer than one might imagine when there are many nodes and edges in the graph. This algorithm can be served as a baseline for possible future works that improve running times and may be useful for implementing do-calculus. They are very close to script languages such as python already, and if possible, showing the results through running the program or providing it as open-source could be even more helpful to scientists who may use this FD strategy for experiment design.
>
> Following your suggestion, we have implemented the algorithm. We have tested over 30 graphs (including the ones from the paper and examples presented to other reviewers) with different parameters, and we verified the results were correct. We plan to release the software as open-source along with the camera-ready paper if the paper is accepted. Thank you for sharing this idea, much appreciated.

---

### Meta-Review · Area_Chair_jzDy · 2022-09-03

**Recommendation:** Accept
**Confidence:** Less certain

**Metareview:**

The paper presents new algorithms for finding and enumerating sets satisfying Pearl's front-door (FD) criterion given a causal diagram. While the reviews have made some helpful suggestions which the authors should implement, they all agree that the paper makes a solid contribution to our understanding of this topic.


**Award:**

No

---

### Decision · Program_Chairs · 2022-09-14

Accept